# A Diffusion Model on Algebraic Varieties with Application to Protein Loop Modeling

## Abstract

The conformation spaces of loop regions in proteins as well as closed kinematic linkages in robotics can be described by systems of polynomial equations, forming real algebraic varieties. These are formulated as the zero sets of polynomial equations constraining the rotor angles in a linkage or macromolecular chain. These spaces are essentially stitched manifolds and contain singularities. Diffusion models have achieved spectacular success in applications in Cartesian space and smooth manifolds but have not been extended to varieties. Here we develop a diffusion model on the underlying variety by utilizing an appropriate Jacobian, whose loss of rank indicates singularities. This allows our method to explore the variety, without encountering singular or infeasible states. We demonstrated the approach on two important protein structure prediction problems: one is prediction of Major Histocompatibility Complex (MHC) peptide interactions, a critical part in the design of neoantigen vaccines, and the other is loop prediction for nanobodies, an important class of drugs. In both, we improve upon the state of the art open source AlphaFold.

## 1 Introduction

Proteins are essential polymeric biological molecules, and knowing the 3D structure of a protein is key for understanding its function. A protein loop is a non-regular contiguous segment of the protein chain which connects the regular structural elements, such as alpha helices and beta sheets, shown in Fig. 1(a). The "non-regularity" of loop structure is expressed as a lack of a fixed periodic pattern of hydrogen bonding typical for alpha helices and beta sheets. This absence of stabilizing hydrogen bonding, compounded by the fact that loops are often located on the surface of the protein and thus exposed to the solvent, makes loop structures more challenging to characterize both experimentally (using X-ray crystallography or Cryo-Electron Microscopy) and computationally in the context of protein structure prediction [Barozet et al. (2021)].

At the same time, protein loops often play key roles in protein function, forming the components of enzymatic sites (such as kinase activation loops, HIV protease flap loops, Dihydrofolate Reductase Met20 loop, etc. [Malabanan et al. (2010)]) as well as serving as the binding sites for other molecules, such as most prominently in Complementarity Determining Regions (CDR) of antibodies [Nowak et al. (2016)]. Such structural motifs appear in many biologically significant contexts. For example, peptide loops are critical components in major histocompatibility complex (MHC)–peptide complexes, where the looped conformation of the peptide enables specific interaction with the MHC binding groove and recognition by T-cell receptors. Additionally, nanobodies – single-domain antibody fragments derived from camelids – possess unique CDR3 loops that are often longer and more structurally diverse than those in conventional antibodies. Understanding the structure and flexibility of such loops is vital for advancing structure-based drug design, immunotherapy, and vaccine development. This discrepancy between functional importance and our limited ability to model them computationally (compared to protein structure in general) makes the problem of predicting the protein loop structures highly relevant, and serves as a motivation for the work presented here.

Protein structure prediction poses a challenge to the scientific community. Recently, progress has been accelerated by AlphaFold (AF) [Jumper et al. (2021); Abramson et al. (2024)], made possible by the advances in the field of machine learning and the availability of data resulting from the decades long accumulation of experimental structures deposited in the Protein Data Bank (PDB)

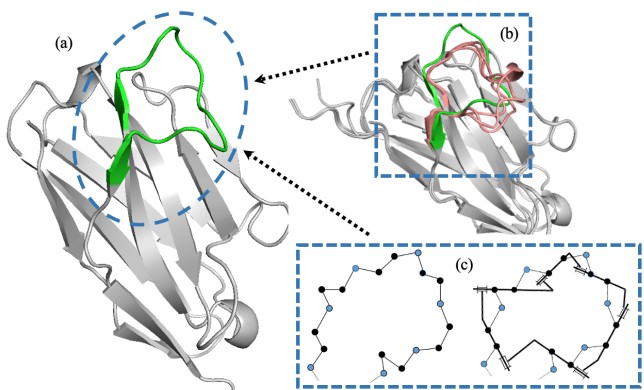

Figure 1: **a:** The experimental structure of the chain of a nanobody (PDB ID: 8SFZ). The circled green segment shows the CDR3 loop region of the nanobody. **b:** The CDR3 loop region in the experimental structure is in green, three predictions from AlphaFold are in red while the rest of the structures are in gray. The predictions of CDR3 loop region are close to each other but not matched with the experimental structure, while other segments are nearly matched. **c:** By considering only the backbone atoms of a loop, we can convert it to a closed 6-revolute kinematic linkage.

[Berman et al. (2009)]. However, even such advanced approaches often have difficulties in predicting certain structural elements, with loops being a prime example (see Fig. 1(b)). These limitations call for the development of novel computational methods.

The general problem of protein structure prediction can be formulated as a generative task of learning the probability distribution $p(\boldsymbol{x}|\boldsymbol{s})$ over protein structures $\boldsymbol{x}$ conditioned on protein sequence $\boldsymbol{s}$ (in the case of partial modeling, like in case of protein loops, we additionally condition the distribution on the non-loop portion of the protein structure $\boldsymbol{r}$ and deal with the target distribution $p(\boldsymbol{x}|\boldsymbol{r}, \boldsymbol{s})$). The idea has received a lot of attention recently, with latent variable generative models, especially diffusion models, being successfully used to generate the structures of proteins and other molecules, including the modeling of protein structures [Watson et al. (2023)], protein backbones [Yim et al. (2023)], small molecules [Xu et al. (2022); Jing et al. (2022)], and molecular docking [Corso et al. (2023)].

The earlier generation of such models does not explicitly incorporate the constraints imposed by chemical bonding and applies the noise to 3D coordinates of each atom independently when constructing the forward process, such as in [Xu et al. (2022); Hoogeboom et al. (2022); Yim et al. (2023); Watson et al. (2023)]. This practice is often referred to as diffusion in Euclidean space and leads to an increased number of denoising steps because all features of the chemical structure have to be learned from the data directly. More recently, a number of approaches have taken advantage of the fact that molecular flexibility is largely limited to the so-called torsional angles formed by a sequence of 4 atoms connected consecutively by three covalent bonds, while bond length and 3-atom bond angles maintain essentially constant values. Representing a molecular structure in terms of torsional angles significantly reduces the dimensionality of the problem and has been recently used to construct more efficient diffusion models for small non-protein molecules [Jing et al. (2022); Corso et al. (2023)].

While the above approaches perform very well for tree-like molecular graphs, the case of protein loops adds additional geometric constraints to the picture, namely the requirement of loop closure: the generated loop structures must have both ends fixed, while the protein chain must remain unbroken, i.e. all chemical bond lengths fall within acceptable margins of error from expected values. The closure condition makes the torsional spaces of closed loops highly constrained subspaces of the hypertori which may involve singularities, and are therefore challenging to learn directly, especially since every loop has a different submanifold from others, determined by its length and the relative position of the two ends. Incorporating the closure constraint into the model as an inductive bias reduces the effective dimensionality of the problem and may produce an architecture which is both more computing- and data-efficient (similarly to how torsional diffusion is more efficient than Euclidean) relative to both torsional diffusion and Euclidean diffusion baselines. Here we propose

such a diffusion model operating on algebraic varieties, which can be used to study the constrained manifold for the loop regions in proteins.

The main contributions of the work are:

1. We propose a diffusion-inspired method suitable for algebraic varieties and applicable to the problem of structure prediction for a broad class of constrained molecules, including protein loops, macrocycles [Jimenez et al. (2023)], stapled peptides [Li et al. (2020)] and MHC-bound peptides.

2. We demonstrate the performance of the architecture on two important biological problems: 1) predicting the structures of peptides bound to MHC receptors and 2) predicting the structures of nanobody CDR3 loops. On an MHC type I dataset containing 78 complexes, we achieve over 15% improvement in terms of median RMSD over the public domain state of the art model AlphaFold 2. Similarly, on a nanobody test set containing 21 cases, the diffusion model improves upon the results of AlphaFold 3 by approximately 13%.

## 2 BACKGROUND AND RELATED WORK

A conformation of a molecule can be represented by coordinates of all atoms in Euclidean space, which can be thought of as an element in $\mathbb{R}^{3N}$, where $N$ is the number of atoms in the molecule. However, the observed variations of bond lengths and angles in experiments are relatively small, and the flexibility of a molecule is mainly determined by the torsional angles at rotatable bonds [Gō & Scheraga (1970); Dinner (2000)]. In the case of proteins, we have chains of amino acids joined by peptide bonds. Each amino acid contributes three atoms to the protein backbone, one Nitrogen and two Carbons, so that a protein of $M$ amino acids $A_i$ entails the backbone of $3M$ atoms, $\{A_i(N_i - C_{\alpha,i} - C_i)\}_{i=1}^M$, and $3M - 2$ rotatable bonds. The peptide bonds $(C_{i-1} - N_i)$ formed by a dehydration reaction between two consecutive amino acids, $A_{i-1}$ and $A_i$, are usually treated as non-rotatable because they tend to have small changes in the structures. Thus, the embedding space of an internal protein backbone conformations is a hypertorus with dimension $2M - 2$. For the purpose of this work, we assume the internal conformation of side chains (short molecular chains branching from the $C_\alpha$ atom and specific for each type of amino acid) to be fixed (but note in passing that as side chain conformations are not subject to closure constraints, they can be straightforwardly handled by using existing approaches based on torsional diffusion, and our architecture can be augmented to include such treatment). In this case, the remaining flexibility in the protein chain comes from the $\phi$ and $\psi$ angles in the backbone (resp. torsions $(C_{i-1}, N_i, C_{\alpha,i}, C_i)$ and $(N_i, C_{\alpha,i}, C_i, N_{i+1})$). Here we focus on the movement of the backbone of a protein loop region which is constrained by its attachment at both ends to the rest of the structure. The backbone of a loop and a schematic of its conversion to a linkage is shown in Fig. 1(c).

To the best of our knowledge, there is no deep learning method to generate loop conformers in algebraic variety space, but several methods have been proposed to explore the conformational space of loops, such as systematic sampling, Molecular Dynamics (MD), Monte Carlo (MC), and geometric methods [Barozet et al. (2021)]. In MD and MC methods, an ensemble of conformations is obtained through computationally intensive simulations. In systematic sampling, with rigid rotor assumption where only the torsional angles are flexible [Gō & Scheraga (1970)], different conformations of the loop can be explored through sampling the backbone torsional angles $\phi, \psi$ with a given granularity. This method is exhaustive and deterministic, but the optimal granularity varies for different molecules. Without considering the constraints at two ends, the generated conformers are usually open. As the structures of molecules can be treated as geometric objects, some geometric methods have also been proposed to sample loop regions, such as Triaxial Loop Closure [Coutsias et al. (2004)] and constrained normal mode analysis (NMA) [López-Blanco et al. (2022)]. In [Coutsias et al. (2004)], the kinematic view of the loop was explored. The fully algebraic method can explicitly account for the closure constraints imposed by having the two ends of the loop fixed. This method was then extended to the KIC method in the Rosetta suite for molecular modeling [Mandell et al. (2009); Stein & Kortemme (2013)]. In [López-Blanco et al. (2022)], constraints of loop closure were added to the regular NMA method to explore the local conformation of loops. However, in such geometric methods, thousands of conformations need to be generated first and the representative conformers can be chosen through clustering based on pairwise root mean square deviation

(RMSD) values. In geometric methods, plenty of redundant conformations will be generated and an energy-based scoring function is still needed to rank the conformations.

Finally, it should be acknowledged that the state of the art in protein structure modeling is currently best exemplified by the AF2 approach [Jumper et al. (2021)] (and the recently made available AlphaFold 3 [Abramson et al. (2024)]), which represents a major breakthrough in the general purpose structural modeling of proteins and sets a new standard for prediction accuracy. Given the prediction accuracy from AF2, in most realistic modeling scenarios, the loop-specific modeling tools will use the predictions made by AF2 or RoseTTAFold [Baek et al. (2021)] as a starting point, refining the predictions of the loop regions while keeping the rest of the structure largely intact. In this context, any improvements that loop-specific modeling tools aim to achieve have to be characterized relative to the predictions of the baseline general-purpose model.

## 3 METHOD: DIFFUSION ON ALGEBRAIC VARIETIES

### 3.1 OVERVIEW

Unlike a free chain, the backbone angles in an $n$-torsion loop are subjected to additional constraints that keep two ends fixed with respect to the rest of the protein. These constraints define an algebraic variety on the hypertorus $\mathbb{T}^n$, which is essentially stitched manifolds possibly featuring singularities. Mathematically, the object of interest is the $(n-6)$-dimensional subvariety of the $n$-Torus defined by a system of trigonometric expressions relating two ends of the loop through a sequence of orthogonal transformations defined by six pivotal rotors along the closed kinematic chain. Expressing all sines and cosines in terms of half-tangents of the six constrained torsions, these trigonometric closure conditions result in a system of polynomials whose real solutions define the alternative conformations of the loop (derivations of the polynomial system can be found e.g. in [Cao et al. (2023)], see also [Angeles (2014), p.375-389]). Standard methods reduce the problem to the solution of a 16-degree polynomial in one of the variables, and from each real root the remaining five variables are determined. The real zero set of the closure polynomial system can have nontrivial topology. This introduces nontrivial variety structure, e.g., the space of a closed canonical octagonal chain is topologically the union of a sphere and a Klein bottle that intersect along two circles [Martin et al. (2010)].

We approach the loop modeling problem by learning a probability distribution $p(\mathbf{x}|\mathbf{r}, \mathbf{s})$ over loop conformations $\mathbf{x}$ conditioned on the remaining protein structure $\mathbf{r}$ and sequence $\mathbf{s}$. For that, we develop a diffusion model operating on algebraic varieties. The details of the chosen diffusion generative models are given in Appendix F. Unlike diffusion on Euclidean spaces or smooth manifolds, diffusion on varieties can be challenging in the vicinity of singularities. We propose a way relying on the tangent space to move as shown in Fig. 2, in the spirit of the Geodesic Random Walk [De Bortoli et al. (2022)]. At each step, the tangential noise is sampled and then the tangent vector is mapped back to the variety to produce a valid step on the variety. We achieve this by applying the $R6B6$ [Cao et al. (2023)] algorithm to maintain loop closure. $R6B6$ (from "6 Rotors/6 Bars") is a robust algorithm to handle loop closure and conformational sampling problem in chains with fixed ends. It uses a system of polynomial equations to solve for the constrained torsions, ensuring the chain remains closed while allowing flexible perturbation of the remaining torsions. In a chain with $n$ flexible torsions, we can select $n-6$ torsions to perturb, and $R6B6$ can be used to solve for the remaining 6 torsions to maintain two ends of the chain fixed with respect to the rest of the protein structure. Details of $R6B6$ is provided in Appendix G. To add proper noise in the diffusion process, we resort to the Jacobian matrix constructed from geometrical loop closure relationships to obtain an orthogonal set of basis vectors for the space of infinitesimal deformations consistent with loop closure. The following sections present the algorithmic details of our method.

### 3.2 DIRECTIONS OF CONCERTED MOVEMENT FOR A LOOP REGION

Consider a loop $\{\mathbf{R}_i\}_{i=1}^n$ with $n \geq 6$ flexible backbone torsions $\zeta_i$, $i = 1, ..., n$, the ends of which are fixed. The torsions $\zeta_i$ are all flexible but should be chosen properly to construct realizations of the loop that are consistent with the closure constraints. The derivation is standard in the robotics literature analyzing systems of revolute joints [Angeles (2014)]. Under certain conditions, we can ensure loop closure by assigning the values for the $n-6$ torsions and solving a system of polyno-

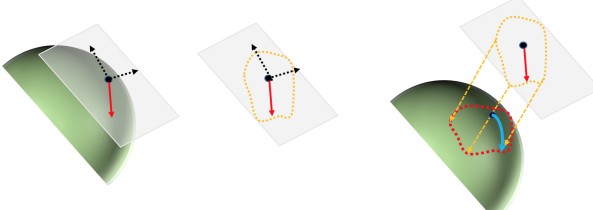

Figure 2: The green denotes the variety and the white plane is the tangent space at the point. The tangential noise is sampled (red line) based on the basis vectors (dashed black lines) of the tangent space. The tangent vector is then mapped back (orange dashed lines with arrows) to produce a valid step on the variety (blue). The orange curved region denotes the boundary of the movement in the tangent space and the red curved region is the boundary of the movement at current state on the variety.

mial equations to determine the remaining 6. Following ideas in [Coutsias et al. (2016)], we select 6 torsion axes (or "Pivots") and divide the chain into seven segments between successive axes. Setting all remaining torsions to prescribed values rigidifies the intervening segments, resulting in the problem of selecting values of the pivot torsions for assembling a revolute chain with fixed ends. Thus the torsional space of the loop is a $(n-6)$-dimensional variety embedded in an $n$-dimensional torus, and the dimension of the tangent space at a regular point is also $n-6$. Given a chain configuration, we may characterize the tangent space directly by considering a concerted change of all torsions $\zeta \to \zeta + d\zeta$ in the loop that keeps its ends fixed. At any atom $\mathbf{R}$ of a chain located forward from the $i$th rotor, the infinitesimal change $d\zeta_i$ at the $i$th torsion will change the position of $\mathbf{R}$ by $\mathbf{\Gamma}_i \times (\mathbf{R} - \mathbf{R}_i)d\zeta_i$, where $\mathbf{\Gamma}_i$ is the unit vector along the $i$th torsional rotation axis and $\mathbf{R}_i$ is the position of $i$th atom. Beyond the fixed ends of the chain, the summation of these changes will be canceled out, so that we have

$$0 = d\mathbf{R} = \sum_{i=1}^{n} \mathbf{\Gamma}_i \times (\mathbf{R} - \mathbf{R}_i)d\zeta_i \Rightarrow \left( \sum_{i=1}^{n} \mathbf{\Gamma}_i d\zeta_i \right) \times \mathbf{R} - \left( \sum_{i=1}^{n} \mathbf{\Gamma}_i \times \mathbf{R}_i d\zeta_i \right) = 0. \quad (1)$$

Since this is true for arbitrary $\mathbf{R}$, past the end of the loop, both expressions in parentheses of Eq. 1 must vanish independently, from which we find

$$\mathbf{P}d\zeta = \sum_{i=1}^{n} \mathbf{P}_i d\zeta_i = 0, \mathbf{P} := (\mathbf{P}_1 \ \mathbf{P}_2 \cdots \mathbf{P}_n) \text{ where } \mathbf{P}_i = \left( \begin{array}{c} \mathbf{\Gamma}_i \\ \mathbf{\Gamma}_i \times \mathbf{R}_i \end{array} \right), \quad (2)$$

where $\mathbf{P}$ is the Jacobian matrix whose dimension is $6 \times n$. The columns of the Jacobian are the Plücker coordinates [Angeles (2014), p.102] of the corresponding axes. The derivation of $\mathbf{P}$ for manipulator linkages can be found in [Hunt (1990), Eq. 11.19], see also [Viquerat et al. (2013)], and the generalized derivation for molecular chains can be found in [Coutsias et al. (2016)]. Basic analysis (the Implicit Function Theorem) guarantees that six of the variables may be expressed as differentiable functions of the remaining ones provided $\mathbf{P}$ has full rank (Theorem A.2.8 in [Sommese & Wampler (2005)]). Given 6 independent columns, the corresponding 6 torsional perturbations can be expressed locally as differentiable functions of the other $n-6$ torsions. Intuitively, the linkage needs 6 DoF to maintain closure, since given the location of one end, placing the other at the correct position and orientation requires, roughly speaking, 3 translational and 3 rotational degrees of freedom. From the singular value decomposition (SVD) of the $6 \times n$ matrix $\mathbf{P}$, we can obtain a set of $n-6$ orthonormal null vectors $\mathbf{v}_i, i = 1, ..., n-6$ with $\mathbf{v}_i \cdot \mathbf{v}_j = 0$ and $||\mathbf{v}_i|| = 1$, forming the basis for the tangent space of the variety at the current point. These vectors in the tangent space provide a set of orthogonal directions for the concerted movement of the torsional angles in the loop. Any infinitesimal perturbation of the loop torsions that can be expressed as a linear combination of the vectors $\mathbf{v}_i$ in the tangent space will keep the loop closed at both ends.

### 3.3 TRAINING AND INFERENCE

We propose a denoising diffusion model to approximate the distribution over loop torsions conditioned on protein sequence and structure. We train a score model $\mathbf{s}_\theta(\boldsymbol{x}_t, t)$ in the tangent space

$span(\{\mathbf{v}_i, i = 1, .., n - 6\})$ of the closure variety, where $\boldsymbol{x}_t$ is the state of geometric graph describing the protein structure at time $t$. The score model therefore predicts a vector $\delta\boldsymbol{\tau}$ living in $span(\{\mathbf{v}_i, i = 1, .., n - 6\})$ that can be expressed in both ambient $n$-dimensional torsional basis and tangential basis $\{\mathbf{v}_i, i = 1, .., n - 6\}$ and is trained to match the score $\nabla_{\boldsymbol{\tau}_t} \log p(\boldsymbol{\tau}_t|\boldsymbol{\tau}_0)$ expressed in tangential basis, where $p(\boldsymbol{\tau}_t|\boldsymbol{\tau}_0)$ is the perturbation kernel of the forward diffusion.

To train the score model, we sample from $p(\boldsymbol{\tau}_t|\boldsymbol{\tau}_0)$ and compute its score. We choose the normal distribution as the kernel for the perturbation samples. The noise scale function is $\sigma_t = \sigma_{\min}^{1-t}\sigma_{\max}^t, t \in [0, 1]$. To add perturbation noise to the torsions of the loop, we first sample $(\tau_1, \tau_2, ..., \tau_{n-6})$ from $p(\boldsymbol{\tau}_t|\boldsymbol{\tau}_0)$, components of the perturbation in tangential basis. The resulting perturbation to the $n$ torsions is given by:

$$\Delta\boldsymbol{\zeta}_t = \tau_1\mathbf{v}_1 + \tau_2\mathbf{v}_2 + .. + \tau_{n-6}\mathbf{v}_{n-6}. \tag{3}$$

However, this tangential perturbation may push us off the variety, breaking the loop. To maintain closure, the $R6B6$ algorithm is applied to map the movement back to the variety (as illustrated in Fig. 2). We select 6 largest components in $\Delta\boldsymbol{\zeta}_t$ and set the corresponding torsions as unknown pivots, after verifying that the corresponding $6 \times 6$ submatrix of the Jacobian is invertible. We next add the remaining $n - 6$ components of $\Delta\boldsymbol{\zeta}_t$ to the corresponding $n - 6$ torsions. The solutions for the 6 pivotal torsions can be solved by $R6B6$, i.e. guarantee that the spatial constraints at two ends of the loop are exactly satisfied. The closest solution $\zeta_i^p$ to the original 6 torsion values $\zeta_i^o$ is selected to have the smallest metric $\sum_{i=1}^{6} |\zeta_i^p - \zeta_i^o|$, where $|\cdot| = 2\pi - |\cdot|$, if $|\cdot| > \pi$. Then the perturbed $n$ torsions are composed of this solution and the $n - 6$ torsions above. The difference $\Delta\boldsymbol{\zeta}_t'$ between the perturbed torsion values and the original values before perturbation provides the noise to the $n$ torsions. In training, if the closure problem is not solvable by $R6B6$ which indicates an infeasible movement, the next perturbation will be tried until one solvable case is sampled successfully. In our experiments, the closure problem can almost always be solved after perturbation only once, with rare failures requiring at most three trials. During training, we sample the time $t$ uniformly and minimize the loss $\mathcal{L}(\theta) = \mathbb{E}_t[\lambda(t)\mathbb{E}[||\boldsymbol{s}_\theta(\boldsymbol{\tau}_t, t) - \nabla_{\boldsymbol{\tau}_t} \log p(\boldsymbol{\tau}_t|\boldsymbol{\tau}_0)||^2]]$, where $\lambda(t) = \mathbb{E}[||\nabla_{\boldsymbol{\tau}_t} \log p(\boldsymbol{\tau}_t|\boldsymbol{\tau}_0)||^2]$ as in [Song et al. (2021); Corso et al. (2023)]. The training procedures are given in Algorithm 1.

---

**Algorithm 1** Training

**Input** Molecular graphs $[G_0, G_1, ..., G_N]$, learning rate $\alpha$
**Output** Score model $\boldsymbol{s}_\theta$
**for** $epoch = 1$ to $epoch_{\max}$ **do**
    **for** $G$ in $[G_0, G_1, ..., G_N]$ **do**
        extract loop region $\mathbf{l}_p$ from $G$;
        compute null vectors $\mathbf{v}_i, i = 1, ..., n - 6$, using Jacobian based on $\mathbf{l}_p$;
        sample $t \in U[0, 1]$;
        set close flag $flag = 0$;
        **while** $flag = 0$ **do**
            sample $\Delta\boldsymbol{\tau}$ from Gaussian $p_{t|0}(\cdot|0)$ with $\sigma_t = \sigma_{\min}^{1-t}\sigma_{\max}^t$;
            $\Delta\boldsymbol{\zeta}_t = \sum \Delta\boldsymbol{\tau}_i \cdot \mathbf{v}_i$;
            $\Delta\boldsymbol{\zeta}_t' = Closure(\Delta\boldsymbol{\zeta}_t)^*$;
            If $\Delta\boldsymbol{\zeta}_t'$ is not None: $flag = 1$;
        apply $\Delta\boldsymbol{\zeta}_t'$ to $\mathbf{l}_p$;
        predict $\delta\boldsymbol{\tau} = \boldsymbol{s}_{\theta,G}(t)^{**}$;
        update $\theta \longleftarrow \theta - \alpha\nabla_\theta||\delta\boldsymbol{\tau} - \nabla_{\Delta\boldsymbol{\tau}} p_{t|0}(\Delta\boldsymbol{\tau}|0)||^2$;
$^*Closure(\Delta\boldsymbol{\zeta}_t)$: 1. select 6 indices with largest components in $\Delta\boldsymbol{\zeta}_t$ as the unknown pivots; 2. apply remaining $n - 6$ of $\Delta\boldsymbol{\zeta}_t$ to $\mathbf{l}_p$; 3. check if there exist solutions to the pivots by using $R6B6$: if true, modify $\Delta\boldsymbol{\zeta}_t$ to $\Delta\boldsymbol{\zeta}_t'$ and return $\Delta\boldsymbol{\zeta}_t'$, otherwise $\Delta\boldsymbol{\zeta}_t' = $ None.
$^{**}\boldsymbol{s}_{\theta,G}(\mathbf{v}_i, t)$ first predicts scalars corresponding to the torsions followed by multiplication with all null vectors $\mathbf{v}_i$.

---

During inference, the null vectors in the current state will be computed by SVD and the tangential score $\delta\boldsymbol{\tau}$ can be predicted from the neural network $\boldsymbol{s}_\theta(\boldsymbol{x}_t, t)$, from which we can obtain the proposed perturbation $\Delta\boldsymbol{\zeta}_t$. The algorithm $R6B6$ is used to check whether the loop is closed after perturbation, i.e. the moved point can be mapped back to the variety. If the loop remains closed, the perturbation $\Delta\boldsymbol{\zeta}_t'$ will be added to the flexible torsions. Otherwise, the step size will be reduced by

10% to see if the reduced perturbation can be added to the structure. The success rate of denoising steps in the inference of our model is around 95%, and it takes approximately 1 second to produce a conformation with 20 denoising steps.

Each denoising step requires the usage of $R6B6$ and SVD. The computational cost of $R6B6$ is around 0.5 ms, while SVD for the $6 \times N$ Jacobian matrix is $O(6N \min(6, N))$, which is linear in $N$. Since $N$ is at most 34 in our use case, this results in a computation time on the order of $10^{-5}$ seconds per step. Given that one diffusion step requires less than 0.1 seconds overall, the cost of SVD is negligible compared to the benefits it provides in efficiently sampling the variety. With all the flexible torsions in the loop updated, we can reconstruct the structure accordingly, which has two ends of the loop closed. The procedures for inference are given in Algorithm 2.

---

**Algorithm 2** Inference

> **Input** Initial molecular graph $G'_0$, number of conformers $K$, number of steps $N$
> **Output** Predicted ensemble $[G'_1, ..., G'_K]$
> **for** $j = 1$ to $K$ **do**
>     extract loop region $\mathbf{l}_p$ from $G'_0$;
>     **for** $n = N$ to 1 **do**
>         set $t = n/N$, $g(t) = \sigma_{\min}^{1-t}\sigma_{\max}^t \sqrt{2 \ln (\sigma_{\max}/\sigma_{\min})}$;
>         compute null vectors $\mathbf{v}_i, i = 1, ..., n-6$, using Jacobian based on $\mathbf{l}_p$;
>         predict $\delta\boldsymbol{\tau} = \boldsymbol{s}_{\theta, G}(t)$;
>         draw $\boldsymbol{z}$ from Gaussian with $\sigma^2 = 1/N$;
>         $\Delta\boldsymbol{\tau} = (g^2(t)/N)\delta\boldsymbol{\tau} + g(t)\boldsymbol{z}$;
>         set close flag $flag = 0$;
>         **while** $flag = 0$ **do**
>             $\Delta\boldsymbol{\zeta}_t = \sum \Delta\boldsymbol{\tau}_i \cdot \mathbf{v}_i$;
>             $\Delta\boldsymbol{\zeta}'_t = Closure(\Delta\boldsymbol{\zeta}_t)$;
>             **if** $\Delta\boldsymbol{\zeta}'_t$ not None **then**
>                 apply $\Delta\boldsymbol{\zeta}'_t$ to $\mathbf{l}_p$, $flag = 1$;
>             **else**
>                 reduce $\Delta\boldsymbol{\tau}$ by 10%;
>     return $G'_j$

---

### 3.4 Architecture of diffusion model

We designed the score model $\boldsymbol{s}(\boldsymbol{x}, \boldsymbol{r}, t)$ to take as input protein structure represented as heterogeneous geometric graph in 3D including all atoms $\boldsymbol{x}$ in the loop region, and a coarse-grained $C_\alpha$ atom representation $\boldsymbol{r}$ of the residues for the remaining fixed part of the protein. Non-loop regions are fixed in the input graph, serve as spatial constraints to guide feasible loop movements while ensuring closure. $C_\alpha$ residue nodes are featurized with one hot amino acid type encoding. All nodes are sparsely connected based on distance cutoffs that depend on the types of nodes being linked and on the diffusion time. To account for roto-translation symmetries inherent to protein structure prediction problem, we use an architecture similar to SE(3)-equivariant Tensor Field Network [Thomas et al. (2018); Geiger & Smidt (2022)] which operates on the molecular geometric graph for interaction layers. The loop atom representations after the final interaction layer are subject to pseudotorque convolution at each rotatable bond. These convolutions produce roto-translation invariant torsional scores for all $n$ rotatable bonds in the loop. The vector of these scores is an $n$-dimensional vector in tangent space $\mathbb{T}_\theta SO(2)^n$ of hypertorus. To account for the closure condition, we orthogonally project the torsional vector onto the tangent space $span(\{\mathbf{v}_i, i = 1, .., n-6\})$ of the closure subvariety at the current point by expressing with $n-6$ coordinates. The resulting $(n-6)$-dimensional vector is treated as the predicted score $\delta\boldsymbol{\tau}$. The basis vectors of this tangent space are obtained through SVD of the Jacobian matrix $\mathbf{P}$ introduced in Eq. 2. More details of the architecture can be found in Appendix D.

While our architecture is similar to SE(3)-equivariant Tensor Field Networks due to their robust handling of geometric symmetries, alternative architectures, such as PointNet [Qi et al. (2017)], could also be considered for processing point cloud data.

## 4 EXPERIMENTS

We evaluated our method on two important problems in protein structure prediction involving loop-like elements: predicting the structures of peptides bound to the Major Histocompatibility Complex (MHC) class I and predicting the structures of nanobody CDR 3 loop (CDR3). The structures were collected from PDB and the SAbDab [Schneider et al. (2022)], respectively, excluding any structures with incomplete loops. We used release time-based criteria to split the dataset and the details are provided in Appendix B.1.

The training was done on experimental structures, and we first validated and tested the model starting from the PDB structures. After confirming that the generated conformations have good diversity, we started our diffusion from AF models to investigate its impact on structure predictions. Specifically, we used AF to predict protein structures from their sequences, and the structures with top confidence served as input for our trained diffusion model. Each structure was split into two parts: the loop region and the remaining part of the protein. The diffusion model then generated conformations for the loop regions. In our diffusion model, a significant hyperparameter is the maximum noise level $\sigma_{\max}$. We set $\sigma_{\min} = \pi/100$ and examined $\sigma_{\max} = \pi/15, \pi/12, \pi/10, \pi/8$. Details of the hyperparameters are provided in Appendix Table 3.

In the experimental results, we use AF as a baseline to compare our method with. The baseline of AF is given in Appendix H. While the main comparison we perform is to the starting structures generated with AF (one per case) as described above, we generate additional structures with AF to provide ensemble-level comparison (which becomes relevant as we generate multiple structures with our approach). It should be mentioned that the outputs from AF are in principle a deterministic function of its inputs, and it has been shown that the inputs can be stochastically subsampled to obtain an arbitrary number of diverse outputs [Del Alamo et al. (2022)]. Additional samples for comparison are generated by relying on this mechanism.

We must point out that we did not train a model to assess the confidence of the generated ensembles. For that purpose, we performed local refinement and scoring of our predictions using AF similar to [Ghani et al. (2021); Roney & Ovchinnikov (2022)]. The generated conformations were then used as a single template for AF and the resulting structures were ranked using CDR3 loop averaged pLDDT. The best structure was selected based on the same metric. To evaluate the results, we computed the backbone RMSD between the refined conformations and the ground-truth loop regions after aligning the protein structures. The RMSD is given in Å, which is a unit of length often used in the field of structural biology and equal to $10^{-8}$ cm.

### 4.1 MHC CLASS I

For the MHC class I dataset, the MHC bound linear peptide has its two ends nearly fixed, while the intermediate part is free to move similarly to a protein loop region. The distribution of peptide lengths in our dataset is given in Appendix B.2. We prepared a subset of 789 structures (with experiment resolution 3.5 Å as a cutoff) involving the peptides of lengths 9 and 10, the most common lengths in the dataset, and initially trained (636), validated (77) and tested (76) the model on this subset. The diversity of the generated ensembles is shown in Appendix C. The trained model was then applied to 78 peptides (released in years 2023 and 2024) with diverse lengths ranging from 8 to 11 residues.

Our predictions started from 1 seed using AF. We then ran 20 trajectories of 20 denoising steps each and used the resulting 20 structures for AF-based refinement and pLDDT scoring. We compared our predictions with those of AF2 and AF3. For AF2, we generated 20 structures with different random seeds. For AF3, we evaluated the five models produced by AF3 server. The results are summarized in Table 1. In general, the prediction of peptides was improved by using diffusion model denoising. When selecting the model with top pLDDT for the peptide out of 20, the median RMSD decreased by 15.8% from 0.95 Å to 0.80 Å compared with AF2, and the mean decreased as well. We also provide RMSD values for AF3 for reference. Details of the values are provided in the left of Fig. 6 in Appendix B.3. One example with large improvement is shown in Fig. 3. Besides RMSD improvement (Left), the peptide model from diffusion also better fits the binding pocket on the MHC (Right), which indicates more potent interactions.

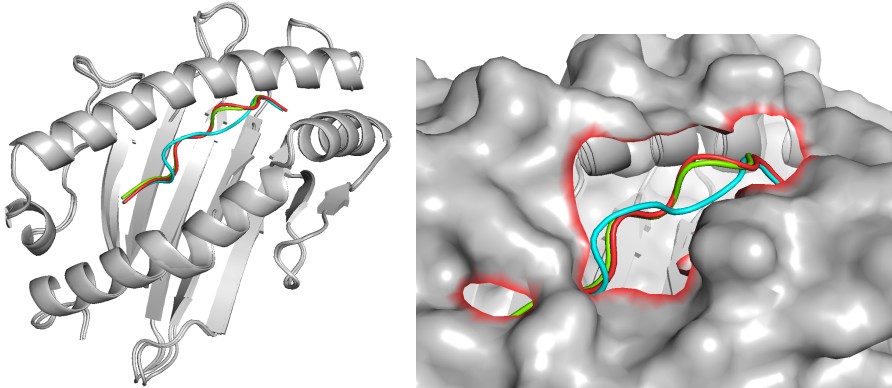

Figure 3: An example of MHC-peptide prediction: PDB ID 8ELG. Left: Peptide from diffusion model (green) has lower RMSD 0.63 Å to PDB structure (red) than AF2 (cyan) with RMSD 1.62 Å. Right: Conformations from diffusion and PDB fit better to the binding pocket on MHC.

Table 1: RMSD comparison for MHC dataset predictions. $^*$ in AF3, the confidence function is different from AF2 and Diffusion.

|  | Top confidence model | | |
|---|---|---|---|
|  | AF2 | AF3$^*$ | Diffusion |
| Mean (Å) | 1.20 | 1.20 | **1.14** |
| Median (Å) | 0.95 | **0.80** | **0.80** |

## 4.2 NANOBODY CDR3 LOOPS

We next examined the performance of our approach on the nanobody CDR3 dataset. The structure of CDR3 loop is critical because it largely determines antigen specificity and binding affinity, making it essential for understanding and engineering nanobody function. Since the CDR3 sequence is known *a priori* [Chothia & Lesk (1987)], the prediction of the CDR3 loop structure becomes the main point of interest, but is significantly complicated by the fact that these loops can be relatively long and hence flexible.

As longer loops pose a more significant challenge and are of high interest, we specifically focused on this class of difficult-to-model systems. Given the structural similarities between nanobodies and antibodies, we enlarged our training set by including antibody loops. By removing the incomplete and repetitive structures, we prepared a dataset of PDB nanobody/antibody structures containing CDR3 loops that are 14 to 16 residues long (732 structures in total with experiment resolution 3.0 Å as a cutoff). The distribution of the lengths of the loops is given in Appendix B.2. The dataset was initially divided into 570 training, 83 validation, and 79 test samples. The model was trained, validated and tested using these structures. The diversity of the generated structures are provided in Table 5 in the Appendix C. In the test set, we have 21 nanobodies with unique sequences after removing the repetitive ones.

For each sequence, the prediction was carried out using AF with 20 random seeds producing 100 conformations. Next the conformation with the top pLDDT in the CDR3 loop was selected from 100. The CDR3 loop of this top confidence conformation was sampled to obtain 100 conformations using the diffusion model with 20 denoising steps. The results are summarized in Table 2. In general, the prediction of CDR3 loops was improved using denoising diffusion model. When selecting the model with top pLDDT for the loop out of 100, the median RMSD decreased from 1.70 Å to 1.48 Å compared to AF3, and the mean also was also smaller. Details of the values are provided in the right of Fig. 6 in Appendix B.3. An example with large improvement is given in Fig. 4. The CDR3 loop is closer to the PDB structure (Left), allowing the interactions between CDR2 loop of the nanobody with the antigen (Right), which will increase the binding potency to the antigen.

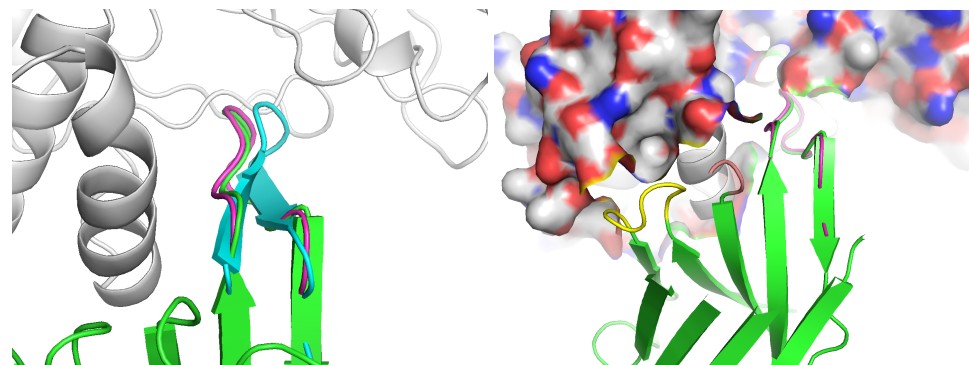

Figure 4: An example of CDR3 loop prediction: PDB ID 8PE1. Left: Antigen and nanobody from PDB are in grey and green respectively, the CDR3 loop model generated by the diffusion model (magenta) has lower RMSD 0.38 Å relative to PDB structure than that by AF3 (cyan) with RMSD 1.42 Å. Right: The overall structure of nanobody-antigen complex interface region.

Although the diffusion model can improve upon AF predictions, we would like to emphasize the importance of diversity in the CDR3 loop conformations. The CDR3 loops of nanobodies are flexible and need to adopt different conformations to interact with different antigens. Depending on the structure of antigens, CDR3 loops can change the conformations when binding to the antigens. This is the place where the diffusion model can provide more help in generating various conformations to predict protein-protein interactions. To demonstrate the diversity of the generated conformations, we provide more analysis on the illustrative example in Fig. 4. The torsion distribution of the amino acids in the generated conformations is an indicator of the diversity. 100 conformations generated from diffusion model before AF refinement and 100 conformations from AF3 are chosen, and the backbone torsions $\phi$ and $\psi$ are calculated. The ensemble generated by the diffusion model has significantly higher diversity than that of AF3. The generated conformations of AF3 only cover limited regions, while diffusion conformations cover larger regions. The torsion distribution for the residue with index 102 in the CDR3 region is given in Fig. 7 in Appendix B.5. Moreover, other quality measures are also calculated. We choose 5 samples from AF3 and 20 samples from diffusion model, and then applied MolProbity server [Williams et al. (2017)] to calculate the MolProbity and Clashscore. The samples from AF3 have excellent but similar MolProbity and Clashscore compared with diffusion samples and PDB structure, while diffusion model generated samples with diverse quality measures and some of them have good MolProbity and Clashscore. The details are given in Appendix B.5. The diversity in structures and quality measures of diffusion samples could help overcome the bias in AF model.

Table 2: RMSD comparison for 21 nanobody CDR3 loops. $^*$ in AF3, the confidence function is different from AF2 and Diffusion.

|  | Top confidence model | | |
|---|---|---|---|
|  | AF2 | AF3$^*$ | Diffusion |
| Mean (Å) | 2.91 | 2.51 | **2.44** |
| Median (Å) | 2.56 | 1.70 | **1.48** |

## 5  CONCLUSION

In this work, we presented a diffusion process on algebraic varieties, which can be applied to generate conformations for protein loop regions with geometrically constrained ends. This is the first diffusion model that can implement loop generation in torsional angle space. The performance of the method was demonstrated using the MHC peptides and the nanobody CDR3 loops. By generating and scoring a few conformations, the model's outputs improve upon the predictions from open source AlphaFold. This model will benefit applications in protein design and drug discovery, as these fields often involve flexible loop regions and long-distance restraints in structures.

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

# A    IMPLEMENTATION DETAILS

**Training.** We used the Adam optimizer [Kingma & Ba (2014)] to train the model. The exponential moving average of the weights during the training and inference steps. Our final score-based diffusion model was trained on a single 48 GB RTX A6000 GPU and Intel Xeon CPU E5-1650 3.6GHz for 300 epochs. It took approximately 45 hours for MHC dataset and 37 hours for nanobody/antibody dataset. We ran 20 denoising steps in the inference, and it took on average 0.8 seconds to generate one conformation.

**Hyperparameters.** To determine the hyperparameters in the diffusion model, we trained smaller models with fewer than 0.5 million parameters before scaling up to the final model (5.5 million parameters). The smaller models were trained for 300 epochs. We used the diversity of the generated conformations with respect to the starting PDB structures to select the hyperparameters. The examined hyperparameters are listed in Table 3. The most important hyperparameter in tuning is the maximum noise level $\sigma_{\max}$. A larger $\sigma_{\max}$ results in large movement during inference, which can break the constrained loop, while a smaller $\sigma_{\max}$ does not introduce noticeable changes in the structures. For the MHC and nanobody/antibody datasets, the models trained with $\sigma_{\max} = \pi/10$ produced the most appropriate results in the loop ensembles, as shown in Appendix C.1. The results of the model trained with this parameter are presented in the main text.

# B    DATASET DETAILS

## B.1    DETAILS OF DATASET SPLITS

We split the MHC class I dataset into three parts: training, validation, and testing. The training dataset, consisting of 636 structures released up to September 30, 2020, and the validation set with 77 structures, released up to February 2, 2022, were used for training the model and optimizing hyperparameters. The test set with 76 structures, containing data released up to August 23, 2023,

Table 3: Hyperparameter options for the score model. The parameter $\sigma_{\max}$, indicating the maximum noise level, was tuned for different datasets. For the MHC and nanobody/antibody datasets, the trained model with $\sigma_{\max} = \pi/10$ produced the best results in the loop ensembles.

| Parameter | Value |
|---|---|
| All atoms for remaining part of protein graph | NO |
| Use Language model embeddings | NO |
| Use hydrogens for ligands | NO |
| Use exponential moving average | YES |
| Maximum number of neighbors in protein graph | 24 |
| Maximum distance of the neighbors | 15 |
| Distance embedding method | Sinusoidal |
| Dropout | 0.1 |
| Learning Rate | 0.001 |
| Activation function | ReLU |
| Convolutional layers | **2**, 4 |
| Number of scalar features | 48 |
| Number of vector features | 10 |
| $\sigma_{\max}$ | $\pi/18, \pi/15, \pi/12, \boldsymbol{\pi/10}, \pi/8$ |

served as the first evaluation of the performance of the trained model. For the nanobody dataset, there are not enough available structures in SAbDab. We then collected an enlarged dataset of nanobody/antibody structures containing 732 data points for initial training, validation and testing. Similarly, we split this dataset into training (570 structures, released up to December 28, 2022), validation (83 structures, released up to February 28, 2024), and testing (79 structures, released up to March 19, 2025).

B.2   DISTRIBUTION OF THE LENGTHS OF LOOPS

The distributions of the loop lengths in MHC peptides and nanobody/antibody CDR3 loops are shown below. In the MHC dataset, only the peptides with length 9 and 10 were used to train the model. In the CDR3 dataset, we used the loops of lengths 14, 15, and 16 to train the model.

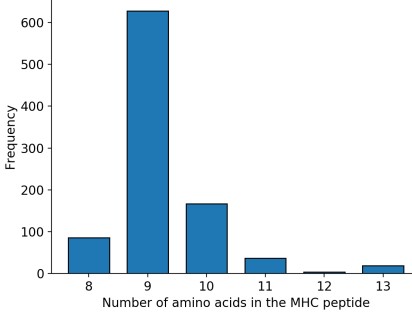 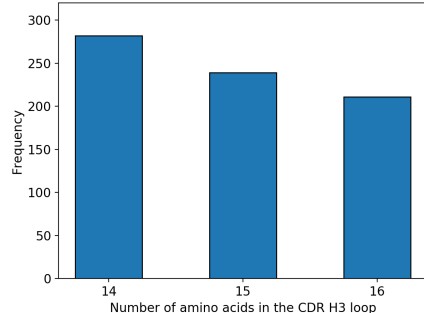

Figure 5: Length distributions for MHC peptides (Left) and nanobody/antibody loops (Right).

B.3   DISTRIBUTION OF RMSD COMPARISON FOR MHC AND CDR3

The overall performance of AF2, AF3, and the diffusion model is given in Table 1 and 2. Fig. 6 provides more details for the comparison.

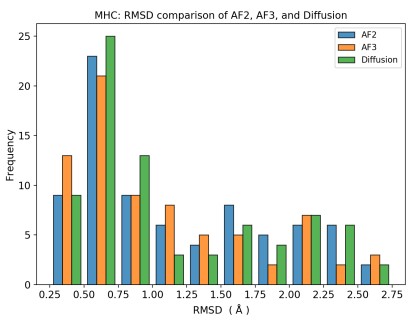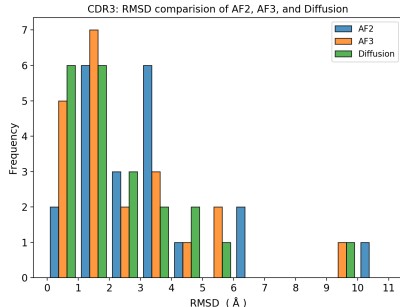

Figure 6: RMSD distributions for the testing of MHC peptides (Left) and nanobodyloops (Right).

### B.4 Performance of top confidence model after removing the similar sequences

For experiments, there exist similarities between the cases in the testing set and training set. For each sequence in the testing set, we checked its similarity between all sequences in the training set. If there exists a sequence in the training set with greater than 50% similarity, we filtered out this case from the testing set, resulting in 44 cases in the MHC dataset and 17 cases in the Nanobody CDR3 dataset. For MHC, we focused on peptides with 9 and 10 because only the lengths of 9 and 10 were included for training. The results after filtering are provided in Table 4.

Table 4: RMSD comparison for MHC (44 cases) and 17 Nanobody CDR3 loops (17 cases) after filtering out the cases with 50% similarity to the training set $^*$ in AF3, the confidence function is different from AF2 and Diffusion.

|  | MHC | | | Nanobody CDR3 | | |
| --- | --- | --- | --- | --- | --- | --- |
|  | AF2 | AF3$^*$ | Diffusion | AF2 | AF3$^*$ | Diffusion |
| Mean (Å) | 1.42 | 1.37 | **1.36** | 3.06 | 2.47 | **2.40** |
| Std. (Å) | 1.39 | **1.24** | **1.24** | 2.86 | 1.42 | **1.41** |

### B.5 Details of torsion distribution, MolProbity, and Clashscore

The torsion distribution of one residue 102 arginine in the loop region of CDR3 of 8PE1 is provided in Fig. 7.

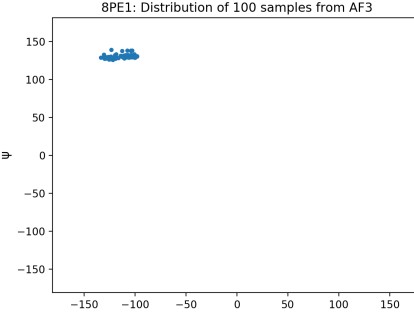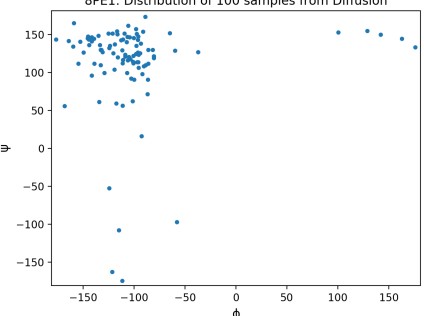

Figure 7: Torsion distribution of one residue for AF3 samples (Left) and diffusion samples (Right).

The MolProbity and Clashscore of the selected 20 samples from AF3, 20 samples from the diffusion model and the PDB structure are shown in Fig. 8.

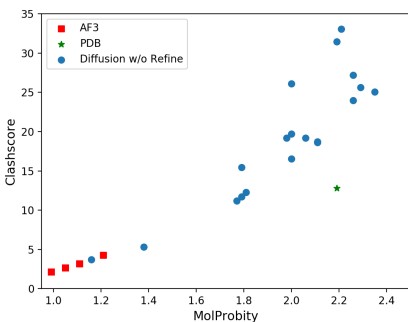

Figure 8: MolProbity and Clashscore for 20 samples of AF3 (in red square) and Diffusion (in blue circle), and PDB (in green star). Note that the values for AF3 are close to each other.

## C ADDITIONAL EXPERIMENTS AND RESULTS

### C.1 IMPACT OF MAXIMUM NOISE SCALE

The maximum noise scale $\sigma_{\max}$ plays a key role in determining the quality of the generated conformations. To evaluate the impact of $\sigma_{\max}$, we examined four different values $\pi, 15, \pi/12, \pi/10, \pi/8$. For each value, we trained and applied the score model to sample the structures in the validation set with 20 denoising steps. The MHC validation dataset has 77 molecules. 20 conformations were generated starting from the PDB structure for every molecule. Then the backbone RMSD values between these generated conformations and the starting conformation were calculated. The mean and standard deviation (std.) of the RMSD values were computed for each molecule. To check the overall diversity in the produced conformations, the mean and std. of these 77 values are provided on the left of Table 5. The mean RMSD value between the AF predictions and the crystal structures is 1.2 Å, so $\sigma_{\max} = \pi/10$ is an appropriate value that produces conformations with 1.13 Å RMSD to the starting conformations on average. Another concern is the success rate of movement in the denoising steps discussed in Appendix E: the larger $\sigma_{\max}$, the lower the success rate will be. We want to maintain a higher success rate and it is below 90% when $\sigma_{\max} = \pi/8$. In the validation set of CDR3 loops, there are 83 molecules and the length of the loop is relatively longer and has more flexibility in the backbone, so we generated 100 conformations for each molecule. The denoising was initiated from the PDB structure for every molecule. Next, the backbone RMSD values between these generated conformations and the starting conformation were computed. The mean and std. of these RMSD values were computed. Finally, the mean and std. for these 83 molecules are provided on the right in Table 5. The mean RMSD values from the prediction of the AF top confidence model to the crystal structure are 4.00 Å (AF2) and 3.81 Å (AF3), so we selected $\sigma_{\max} = \pi/10$ as an appropriate value from which the generated ensembles have an average RMSD value around 4.00 Å to the starting conformation.

Table 5: Average RMSD values of generated ensembles in the validation set with different $\sigma_{\max}$ for MHC (left) and Nanobody/Antibody (right).

|          | MHC |  |  |  | Nanobody/Antibody |  |  |  |
|----------|-----------|------------|-------------|-----------|-----------|------------|-------------|-----------|
|          | $\pi/15$ | $\pi/12$ | $\boldsymbol{\pi/10}$ | $\pi/8$ | $\pi/15$ | $\pi/12$ | $\boldsymbol{\pi/10}$ | $\pi/8$ |
| Mean (Å) | 0.88 | 1.03 | 1.13 | 1.30 | 3.02 | 3.55 | 4.03 | 4.60 |
| Std. (Å) | 0.61 | 0.65 | 0.67 | 0.67 | 2.32 | 2.43 | 2.45 | 2.52 |

### C.2 DENOISING STEPS

Another important parameter is the number of denoising steps. We examined 4 different values $5, 10, 20, 40$. For each value, we applied the score model trained with $\sigma_{\max} = \pi/10$ to the structures in the validation sets for both MHC and nanobody/antibody. The similar RMSD values given in

Appendix C.1 were computed for different ensembles provided in Table 6. The model with 20 denoising steps achieves the most appropriate performance.

Table 6: Average RMSD values of the generated ensembles in the validation set with different denoising steps for MHC (left) and Nanobody/Antibody (right).

|          | MHC |    |    |    | Nanobody/Antibody |    |    |    |
|----------|------|------|--------|------|------|------|--------|------|
|          | 5 | 10 | **20** | 40 | 5 | 10 | **20** | 40 |
| Mean (Å) | 0.93 | 1.02 | 1.13 | 1.32 | 3.51 | 3.74 | 4.03 | 4.45 |
| Std. (Å) | 0.54 | 0.59 | 0.67 | 0.74 | 2.13 | 2.23 | 2.45 | 2.64 |

## C.3 SEVERAL RUNS COMPARISON

To test the consistency of the model, we implemented three tests for each test set. The experiments were implemented with $\sigma_{\max} = \pi/10$ and 20 denoising steps, starting from the PDB structures in the test sets. The similar RMSD values given in Appendix C.1 were computed for different ensembles provided in Table 7. For both datasets, the model can achieve consistent results based on the average RMSD values criteria in the test sets.

Table 7: Average RMSD values of the generated ensembles in the test set from three experiments for MHC (left) and Nanobody/Antibody (right).

|          | MHC |    |    | Nanobody/Antibody |    |    |
|----------|--------|--------|--------|--------|--------|--------|
|          | Test 1 | Test 2 | Test 3 | Test 1 | Test 2 | Test 3 |
| Mean (Å) | 1.22 | 1.25 | 1.24 | 3.86 | 3.91 | 3.87 |
| Std. (Å) | 0.66 | 0.63 | 0.67 | 2.37 | 2.42 | 2.37 |

## D COMPARISON BETWEEN REGULAR DIFFUSION IN TORSIONAL SPACE AND DIFFUSION ON ALGEBRAIC VARIETIES

In the diffusion on algebraic varieties, the basis vectors for the tangent space at a point on the variety depend on the position of that point. Therefore, we have to compute the basis vectors at every step (both in the forward and denoising processes) using the Jacobian matrix $\mathbf{P}$ in Eq. 2 and SVD. The differences between diffusion in torsional space of DiffDock [Corso et al. (2023)] and diffusion on algebraic varieties for the structure with $n$ flexible torsions are summarized in Table 8. In the Diff-Dock framework, the heterograph contains both the receptor and the ligand, and the neural network predicts the binding poses of the small molecule to the protein by modeling translation, rotation, and torsional changes in the ligand. The score model consists of several layers, including embedding layers, interaction layers, and a pseudoscalar layer. The diffusion on algebraic varieties is developed in a similar way. The graphs are constructed using detailed atomic representation for the loop and a coarse-grained representation for the rest of the protein. Moreover, the only flexibility in the loop arises from the backbone torsions $\phi$ and $\psi$ to maintain the loop closure constraints, excluding the translation and rotation of the loop region. A comparison of the neural network structures for diffusion on algebraic varieties and in the torsional space of DiffDock is shown in Fig. 9.

Table 8: Comparison between diffusion on algebraic varieties and diffusion in torsional space for the chain with $n$ flexible torsions.

|                                 | algebraic varieties             | Torsional space               |
|---------------------------------|---------------------------------|-------------------------------|
| Dimension of tangent space      | $n-6$                           | $n$                           |
| Basis vectors of tangent space  | Null vectors of Jacobian matrix | Standard basis of $\mathbb{R}^n$ |
| Map function                    | $R6B6$                          | Exponential map               |

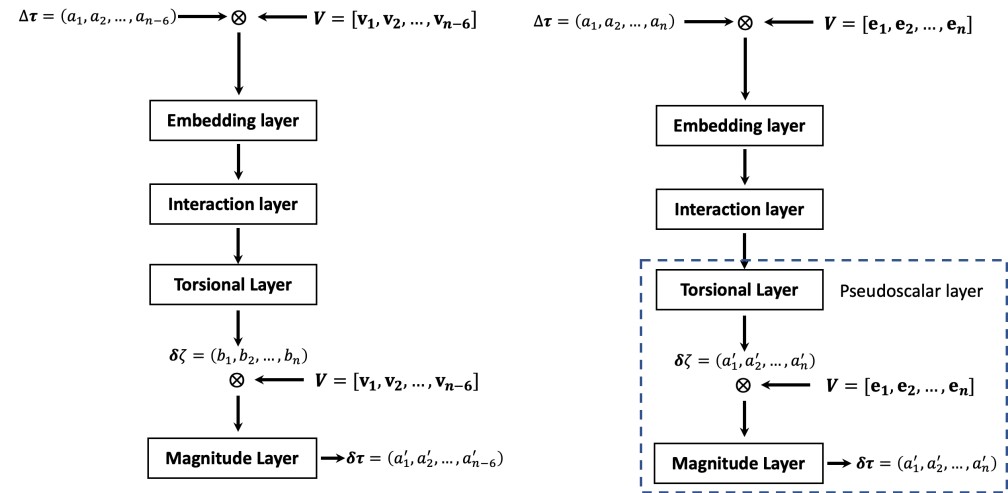

Figure 9: Schematic processes of training for diffusion on algebraic varieties (left) and in the torsional space of DiffDock (right). The sampled noise $\Delta\tau$ is multiplied by the basis vectors $\mathbf{V}$ and then applied to the structure. The structure is next input to the embedding layer, interaction layer, and torsional layer in sequence. The torsional layer outputs scalars $\delta\zeta$ for the corresponding $n$ torsions. The vector $\delta\zeta$ are multiplied with the basis vectors $\mathbf{V}$ in the magnitude layer to predict the magnitudes $\delta\tau$ on each direction of the vectors. In DiffDock, the combination of the torsional and magnitude layers compose the pseudoscalar layer because the standard basis vectors $[\mathbf{e}_1, \mathbf{e}_2, ..., \mathbf{e}_n]$ can be chosen for $\mathbf{V}$ and $\delta\tau = \delta\zeta$, while diffusion on algebraic varieties needs the Jacobian from Eq. 2 to obtain $[\mathbf{v}_1, \mathbf{v}_2, ..., \mathbf{v}_{n-6}]$.

# E  LIMITATIONS OF THE MODEL

The diffusion model produces various conformations for the loop regions and improves the accuracy of structure predictions, as demonstrated in Section 4. However, the diffusion model also has limitations as discussed below.

The first limitation is the removal of flexibility in the side chains from the modeling. In the diffusion model, the initial side chain conformations are used throughout the denoising steps, so they may clash with other parts of the protein, especially when the side chain is long, such as in Lysine and Arginine. The modeling of side chains is a challenging work and there exist backbone-dependent rotamer libraries, such as the one by [Dunbrack Jr & Karplus (1993)], to produce good side chain conformations. We leave the exploration of the side chains to future work in a way that would leverage existing diffusion models to account for the flexibility in the side chains.

Another limitation lies in the rate of successful movement in the denoising steps. When training the model, in every epoch, one structure will try to move in the randomly sampled direction until the movement succeeds. In practice, it usually takes only one trial and rarely more than two trials. On the other hand, in the denoising steps, the movement of some steps would fail which means the moved point cannot be mapped back to the algebraic variety. In such a case, the step size was reduced by 10%. If it still fails, then the movement will shrink for another 10% until the movement succeeds. In the experiments, the success rate of denoising steps is around 95%, so the reduced steps will not have much impact on the generated structures. The failures usually occur near singularity points by crossing loci where the solutions merge and disappear. We can detect their proximity for a given choice of 6 pivots by the fact that the $6 \times 6$ submatrix of the matrix $\mathbf{P}$ in Eq. 2 becomes nearly singular.

Another limitation is in the choice of the 6 pivots when mapping the perturbed points from the tangent space back to the variety. We select the indices based on the magnitude of components in the perturbation, and the 6 indices corresponding to the largest magnitudes are chosen as pivots. The goal is to minimize the effect of the map on the the direction of the perturbation. Suppose the proposed perturbation in the tangent space is $\zeta_i^p$ and the mapped perturbation is $\zeta_i^m$. In our

experiments, the difference between them $|\zeta_i^p - \zeta_i^m|$ is at the scale 1e-2. By choosing the largest components in the perturbation, the ratio of the difference over the proposed perturbation $|\zeta_i^p - \zeta_i^m|/|\zeta_i^p|$ is minimal. This criterion might not be optimal, and the optimal choice is an open problem that will be addressed in our future work. Moreover, to examine the relationship between failure and the selected submatrix, we did another test for one example 8PE1 with 20 denoising steps and generated 100 conformations. In total, there are 2000 steps and the number of failure steps is 82. For these cases, the plot of singular values of the $6 \times 6$ submatrix is shown in the left of Fig. 10. Some of the smallest singular values are at the scale of 1e-3. The determinant of the corresponding $6 \times 6$ submatrix can be at the scale of 1e-4. As they are close to the singular points, the matrix might vanish quadratically and the singularity is not easy to be detected by the first order Jacobian, and special design is needed to study the singularity. Design of special examples and exploration of the variety near singular points will be our future research work. We next examined the step index of the failure cases shown in the right of Fig. 10. In every denoising process, there are 20 steps in total with indexing from 0 to 19 in which 0 is the first step. The majority of the failures occur in the first 10 steps, which indicates that a larger step size is more likely to introduce failures.

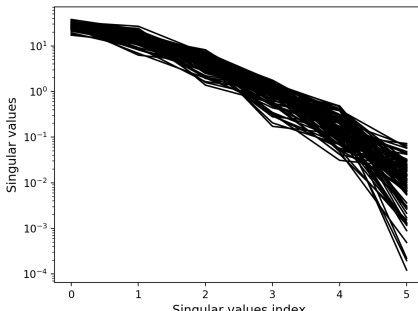 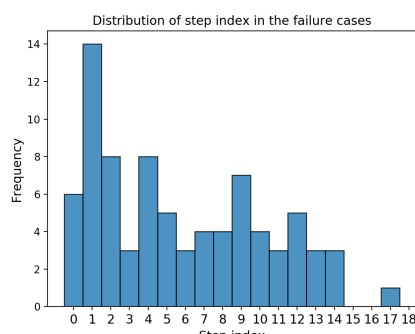

Figure 10: Left: Singular value plot for 82 failure cases in 8PE1 testing, x-axis denotes the index starting from 0, y-axis denotes the semilog of singular values. Right: Distribution of the index of the denoising steps in the failure cases.

## F  DIFFUSION GENERATIVE MODEL DETAILS

Consider the starting distribution $p_0(\boldsymbol{x})$ of a forward diffusion process described by a stochastic differential equation:
$$d\boldsymbol{x} = f(\boldsymbol{x}, t)dt + g(t)d\boldsymbol{w}, t \in (0, T)$$
where $\boldsymbol{w}$ is the Wiener process, $f(\boldsymbol{x}, t) = 0$ and $g(t) = \sqrt{d\sigma^2(t)/dt}$ with $\sigma_t = \sigma_{\min}^{1-t}\sigma_{\max}^t, t \in [0, 1]$, and $\sigma_{\min} = \pi/100$. Due to the compactness of the variety, the prior distribution $p_T(\boldsymbol{x})$ is a uniform distribution.

In the reverse process, we have

$$d\boldsymbol{x} = [-g^2(t)\nabla_{\boldsymbol{x}} \log p_t(\boldsymbol{x})]dt + g(t)d\bar{\boldsymbol{w}}. \tag{4}$$

The score based generative model learns the score of the marginal distribution $\nabla_{\boldsymbol{x}} \log p_t(\boldsymbol{x})$ at $t$, through which we can derive the reverse process with Eq. 4 and generate samples from $p_0(\boldsymbol{x})$.

## G  R6B6 DETAILS

Mapping the tangential noise back to the variety without breaking the loop closure conditions was done using $R6B6$ introduced in [Cao et al. (2023)]. As demonstrated in [Cao et al. (2023)], the algorithm is more stable and accurate than existing robotic algorithms for the inverse kinematics (IK) problem in mechanism theory. The IK problem is important in manipulator linkages and was first solved by a fully algebraic method of optimal degree 16 in [Lee & Liang (1988)], which has applications in robotic arms [Angeles (2014)] and molecular chains [Kolodny et al. (2005)]. In

controlling a robotic arm, the IK goal is to find all solutions for the rotatable torsional angles in the chain to put the hand at a desired position, which is similar to finding all solutions for torsional angles in a molecular chain to satisfy closure constraints. We will show how the algorithm can be applied to the backbone of a loop discussed in our work here and provide a brief description of the algorithm.

The basic question in our problem is: in the backbone of a loop region containing $n$ flexible torsions, given other $n - 6$ torsions, can we find solutions to the 6 pivotal torsions to maintain the closure conditions at two ends of the loop? The backbone of a loop region can be converted from a long chain to a closed 6-revolute (6R) chain, as shown in Fig. 1(c). This conversion is similar to the transformation from a cyclic molecule to a closed 6R chain in [Cao et al. (2023)]. Specifically, in the backbone of a loop, suppose the Cartesian coordinates of $N$ atoms are $a_i = (x_i, y_i, z_i)$, $i = 1, ..., N$. By making a virtual bond between the $N$th and the 1st atom, we can obtain a closed chain. The bond lengths, bond angles, and torsion angles are then computed based on the coordinates of consecutive atoms. The indices for the bonds are from 1 to $N$ with $(a_i, a_{i+1})$ being the $i$th bond, $(a_{i-1}, a_i, a_{i+1})$ forming the $i$th angle, and $(a_{i-1}, a_i, a_{i+1}, a_{i+2})$ corresponding to the $i$th torsional angle. To use $R6B6$, we need indices for the 6 pivotal rotors, $i_j, j = 1, .., 6$. The inputs are summarized in Table 9.

| | Parameters (dimension) |
| --- | --- |
| $N$: | number of atoms |
| $i_j$: | selected axis as the pivots (6) |
| $b_{\text{len}}$: | lengths of the bond between consecutive atoms ($N$) |
| $b_{\text{ang}}$: | angles between consecutive bonds ($N$) |
| $t_{\text{ang}}$: | torsional angles at every bond ($N$) |

Table 9: Details of the input parameters in the backbone of a loop.

However, in the protein backbone of our case, the peptide bonds were treated as fixed (although they may be set to cis or trans form if so desired), so the choice of the 6 pivots will skip the indices for peptide bonds and the last indices for the virtual bond. Suppose there are $n$ flexible torsions. When adding perturbation to the $n$ flexible torsions, we first select six pivots for the six largest components in the noise. Next the $n - 6$ dimension torsional noise is added to the $n - 6$ non-pivotal torsions. $R6B6$ is next used to solve for the 6 pivotal torsions. After selecting six pivots, the segment between two consecutive pivots will be replaced by a virtual bond so we can convert the chain into a closed chain with 6 pivotal bonds and 6 virtual bonds. Next, the common normal between two consecutive pivots will be computed, resulting in a revolute joint in manipulator linkages where the consecutive bonds are vertical to each other. A revolute joint has one degree of freedom at the rotatable torsional angle, which is similar to the flexible bond in molecular chains. After calculating all 6 common normals, we have a closed 6R linkage with at most twelve orthogonal axes.

In the IK problem, a system of polynomial equations [Angeles (2014), Eq.(9.23ab); Cao et al. (2023), Table 1] can first be formulated based on the closure conditions. By using Denavit-Hartenberg coordinates [Denavit & Hartenberg (1955)], we can then convert the system to a set of polynomials in terms of the 6 unknown torsional angles at the joints. By using singular value decomposition, the system can be reduced to a smaller system containing 3 unknowns. Based on the smaller system, a matrix pencil of size $16 \times 16$ is formulated by a well posed algebraic process. These 3 unknowns can be obtained by solving the $16 \times 16$ matrix pencil which is a generalized eigenvalue problem. One of the torsions corresponds to the real eigenvalues, and the other two can be obtained from the components in the corresponding eigenvector. Once these three are found, the remaining three torsions can be computed easily. There are possibly up to 16 alternative solutions and each solution results in a different configuration. These solutions are in the transformed 6R system, so they will be further transformed back to the values in the original molecular chain.

Suppose $R6B6$ gives $L$ solutions to the problem which means there exist solutions for the pivotal torsions to maintain the closure conditions. In our work here, we will use the following criteria to select the closest solution to determine the movement in the denoising steps: $\min_{k=1}^{L} S_k$, where $S_k := \sum_{j=1}^{6} |\zeta_j^k - \zeta_j^o|_C$. $\zeta_j^k$ is $j$th component in the $k$th solution, $\zeta_j^o$ is original torsion value, and $|\cdot|_C = 2\pi - |\cdot|$, if $|\cdot| > \pi$. The whole procedure is summarized in Algorithm 3.

---

**Algorithm 3** Mapping by R6B6

---

1: **Input** A loop of $N$ atoms, and tangential perturbation $\Delta\boldsymbol{\zeta}_t$ for $n$ flexible torsions
2: **Output** Mapped perturbation $\Delta\boldsymbol{\zeta}_t'$ to $n$ flexible torsions
3: Construct a closed ring by connecting a virtual bond in the loop;
4: Formulate a dictionary of indices between $N$ atoms and $n$ flexible torsions;
5: **for** $i = 1$ to $N$ **do**
6:     Compute parameters for bond lengths $b_{len}$, bond angles $b_{ang}$, torsional angles $t_{ang}$;
7: Choose pivots $i_j$; $j = 1, \ldots, 6$ based on $\Delta\boldsymbol{\zeta}_t$; let $\zeta_j^o := t_{ang,i_j}$;
8: Add perturbation in $\Delta\boldsymbol{\zeta}_t$ to the non-pivotal $n-6$ torsions and obtain $t_{ang}'$;
9: $[L, \boldsymbol{\zeta}] = R6B6(b_{len}, b_{ang}, t_{ang}', i_j)$, $L$ is number of solutions, $\boldsymbol{\zeta} = \{\zeta^k\}$ are solutions;
10: Choose one solution $\zeta^{\min}$ by $\min_{k=1}^{L} S_k$, where $S_k = \sum_{j=1}^{6} |\zeta_j^k - \zeta_j^o|_C$;
11: Set perturbation at pivots $i_j$ as $\Delta\boldsymbol{\zeta}_t^p = \zeta_j^{\min} - \zeta_j^o$;
12: Replace perturbation $\Delta\boldsymbol{\zeta}_t$ at pivots $i_j$ by $\Delta\boldsymbol{\zeta}_t^p$ and return $\Delta\boldsymbol{\zeta}_t'$;

---

# H  ALPHAFOLD2/3 BASELINES

For comparison with the baseline AlphaFold2 (AF2) and AlphaFold3 (AF3) methods, we generated the same number of conformations for the baselines as the diffusion-based protocol: 20 from AF2 against 20 from diffusion model for MHC I, 100 from AF3 against 100 from diffusion model in CDR3 loops. For AlphaFold2, we used the ColabFold implementation of AlphaFold2 with ColabDB/MMseqs Multiple sequence alignments (MSAs), and ran each of the 5 standard parameter sets with either 4 (MHC I) or 20 (CDR3 loops) random seeds, for a total of 20 or 100 samples, respectively. Similarly, we ran the standard implementation of AlphaFold3 using standard MSAs to generate a total of 20 (MHC I) or 100 (CDR3 Loops) samples using different random seeds. Dropout was not used.

