# OpenReview forum: "A diffusion model on toric varieties with application to protein loop modeling"
_ICLR.cc/2026/Conference — Submitted to ICLR 2026_

### Official Review · Reviewer_mnqi · 2025-10-28

**Soundness:** 2
**Presentation:** 2
**Contribution:** 2
**Rating:** 2
**Confidence:** 3

**Summary:**

This paper presents a novel generative model for protein loop modeling, a challenging task due to the geometric constraints imposed by fixed loop start and end points. The authors frame this problem as learning a distribution on a toric variety, which is the manifold of valid loop conformations.

The core contribution is a diffusion model that operates directly on this constrained manifold. Instead of diffusing in Euclidean space and learning the constraints, the model diffuses in the local tangent space of the variety. This tangent space is elegantly defined as the (n-6)-dimensional null space of the loop closure Jacobian matrix. A score model is trained to predict noise in this tangent space. To ensure the denoising steps remain on the manifold (i.e., the loop remains closed), the authors employ the R6B6 algorithm as a projection function to map predicted movements back onto the variety.

The method is evaluated on two biological problems: modeling MHC-bound peptides and nanobody CDR3 loops. On both datasets, the proposed model, when used to refine AlphaFold predictions, achieves a lower mean and median backbone RMSD than the open-source AlphaFold 2 and AlphaFold 3.

**Strengths:**

The paper’s primary strength is its novel formulation of constrained generative modeling. Applying diffusion models to complex, non-Euclidean manifolds such as toric varieties is nontrivial and valuable. Using the Jacobian’s null space as the local tangent space for diffusion is mathematically sound and elegantly builds kinematic constraints into the model architecture.

**Weaknesses:**

- Methodological Opacity and Questionable Robustness: The R6B6 algorithm is central to the method: it is the projection function that enforces manifold constraints. Yet it is treated as a black box and merely cited. A reader cannot understand or reimplement the core mechanism without mathematical detail. The authors should provide a mathematical description of the R6B6 algorithm and the toric variety used here. If they prefer to keep this as an empirical paper, they must compensate with additional experiments demonstrating robustness.

- Limited Scope and Generalization Concerns: The model is trained on very small datasets (N = 636 for MHC, N = 570 for nanobody/antibody) and tested on similarly small sets (N = 78, N = 21). The nanobody training set is restricted to lengths 14–16. These facts raise concerns about overfitting and generalization. It is unclear whether the complex, specialized model learned a generalizable manifold-diffusion process or largely memorized the conformational landscape of its small training sets.

**Questions:**

1. To substantiate the claims in Tables 1 and 2, please provide histograms comparing the full RMSD distributions for all generated samples from your model, AF2, and AF3. Summary statistics alone are not sufficient to assess distributional differences.

2. When computing the SVD of the closure Jacobian, is there a clear elbow at rank 6? Please show singular-value plots so readers can judge whether a meaningful null space exists or whether the null space is effectively trivial.

3. In Algorithm 1, the Closure function selects six pivots for the R6B6 solver based on the “largest components in $\Delta\zeta_t$". What is the theoretical or empirical justification for this heuristic? How sensitive is the reported 95% success rate to this pivot-selection strategy?

4. I do not think AlphaFold alone is a sufficient baseline. The authors should provide ablations, such as a diffusion model trained on the same small datasets in Euclidean space (with AF-based refinement and pLDDT scoring), or a non-diffusion model constrained to the toric variety. These comparisons would help justify the need for manifold constraints or a diffusion model.

---

> ### Author Response · Authors · 2025-11-24
> **Part I: Discussions of weaknesses and answers to Question 1-3**
>
> Answer to weakness 1: Thanks for pointing this out. To avoid interrupting the indexing of the sections, we provided details about the usage of R6B6 in Appendix G and modified the description in the main text accordingly.
>
> Discussion of weakness 2: We agree with the reviewer that the dataset is rather small. However, we want to highlight the fact that it is this low data availability that motivated us to develop this learning architecture. Our goal here was to create an algorithm which is less demanding in terms of training data by introducing structural regularization in the form of geometric restraints inherent to the closed loop structures directly into the model. We hope that the small, but significant improvement we achieved despite data limitations demonstrates the validity of the approach. To address  the issue of potential overfitting, we have analyzed the similarity of the test data to the training dataset and report the results for a subset of the testing set in Appendix B.4. For each sequence in the testing set, we checked its similarity between all sequences in the training set. If there exists a sequence in the training set with greater than 50% similarity, then we filtered out this case from the testing set, resulting in 44 cases in the MHC dataset and 17 cases in the Nanobody CDR3 dataset. The improvement of our diffusion model over the AF model persists. We decided to include our initial results in the main text because they are more complete to show the performance of the models.
>
> Following are the answers to the questions.
>
> Question 1 Answer: Thanks for the suggestion, we added the distribution for all cases as histogram in the Appendix B.3.
>
> Question 2 Answer: As we discussed, the rank is always 6 in our examples because they are all molecular structures from experiments or structure predictions. The elbow is a typical place where singularity manifests and is very interesting. We feel the elbow singularity as a typical fold bifurcation is quadratic or higher-order degeneracy, so it is hard to detect its proximity and overshoot. Another possible form of singularity is when two solution branches cross (as happens, e.g. in cyclooctane), but we did not encounter such cases either in the examples we studied. The exploration of singular cases is very interesting, and we will work on it in the future through the design and study of specific examples. We ran more tests on one example (8PE1) with 20 denoising steps and 100 conformations. In these 2000 steps, there are 82 failure cases. We added a singular value plot for these failure cases in Appendix E. In the failures, the smallest singular value is at the scale 1e-3. The determinant of the 6x6 submatrix is at the scale 1e-5. The step size might be large so the proposed movement crosses the elbow singular points. We will design more experiments for the detection of proximity to singularity in the future.
>
> Question 3 Answer: In principle, any 6 can be chosen in the set. But we need a consistent way to select the pivots. We want to reduce the impact of the map to the proposed noise. If we randomly select pivots at every step, we will introduce another layer of randomness to the model, which might not use the information learned from the training properly, so we do not have a guarantee to use the information we have learned from training. If we see the perturbation as a vector of movement, we do not want to change the direction of the vector too much, which may distort the proposed movement direction, so we choose the largest components.  Had we instead chosen to use as pivots the variable exhibiting the smallest change, and used as new reference coordinates the rest including the variables exhibiting the largest change, then the mapping onto the variety would likely involve larger changes in the pivots as well, resulting in larger steps and lower accuracy. To resolve the concern regarding the success rate, we did a quick test on one structure (100 conformations with 20 denoising steps) using a trained model which is using the 6 largest components in training. In the sampling, 6 smallest components were chosen. It showed that the success rate is also greater than 95%. However, the goal of this test was to check the success rate with different pivot selections and generated samples were not examined. We used a criterion to balance change and feasibility in current work and the exploration of the optimal selections will be future work.

---

> ### Author Response · Authors · 2025-11-24
> **Part II: Answer to Question 4**
>
> Question 4 Answer: We do agree with the reviewer’s excellent suggestion that a small Euclidean space diffusion model taken as a baseline would provide a better demonstration of data advantages of loop-constrained approach presented here, and it would be a good idea for future work. However, we argue that the advantage of our approach over the state-of-the-art AF2 and AF3 models which we have demonstrated in the manuscript is potentially a stronger result in absolute terms as these models had much larger training datasets and parameter counts.  In fact, AF3 is already using a Euclidean diffusion module for the whole structure to improve the prediction accuracy, and it is unlikely that the hypothetical small Euclidian space model would reach the same accuracy. In this sense, we compared our method with a larger Euclidean space diffusion model (which has AF-based refinement and pLDDT scoring). On the other hand, traditional methods use  a stochastic process (or Monte Carlo or a similar idea) to do sampling, which requires a large number of attempts to get closed loop conformations. In [Jing et al. 2022], the authors compared their torsional diffusion model for small molecules with RDKit, which is a traditional method using a stochastic process. As can be expected, the diffusion model achieved much better performance given the same number of generated conformations. We provided a distribution of one pair of backbone torsion angles for one example in Fig. 7 in Appendix B.5 to demonstrate the diversity in the generated conformations. We think the current comparison is good enough to show that the diversity generated by the diffusion model for the loops provides a good way to overcome the bias in AF3. Moreover, this diffusion method can be applied to other constrained problems.

---

> > ### Comment · Reviewer_mnqi · 2025-11-26
> > **Response to Authors**
> >
> > Thank you for your clarification. I still have some follow-up questions:
> > 1. From Figure 6, I do not think you can really argue that "the prediction of peptides was improved by using diffusion model denoising." The variance of the RMSD is large. Can you actually reject the null hypothesis statistically?
> > 2. Regarding the R6B6 algorithm: To clarify, I wanted a written algorithmic description using your mathematical notation, in a structured way like Algorithm 1. I suspect some details were adjusted for your specific case. Could you provide it?

---

> > > ### Author Response · Authors · 2025-12-04
> > > **Response to Follow-Up Questions 1–2**
> > >
> > > 1. Thanks for the question. All the values are computed by the comparison to experimental structures, not by sampling from a distribution. We do not think the null hypothesis test is a good way to show the improvement. We have to admit that we cannot reject the null hypothesis. Even the test for AF2 and AF3 cannot reject null hypothesis. In terms of the sentence mentioned in the question, we think it is for the MHC dataset. In this dataset, we generated conformations from AF2, and then applied diffusion based on that and the final results improved upon AF2. Without any information about AF3, we achieved comparable results from AF3 and we provided the results from AF3 for references. Moreover, in the nanobody dataset, we generated conformations from AF3 and applied diffusion based on those, and the final results improved upon AF3.
> > >
> > > 2. We have added more description and pseudocode in Algorithm 3 in Appendix G. You are right, some details were adjusted for our case. Specifically, we added a virtual bond between the last and the first atoms to form a closed ring and the choice of pivots was also adjusted to our case because the flexible bonds in the backbone are not always consecutive and peptide bonds are fixed in our treatment.

---

### Official Review · Reviewer_p11k · 2025-10-31

**Soundness:** 3
**Presentation:** 4
**Contribution:** 3
**Rating:** 4
**Confidence:** 3

**Summary:**

This paper introduces a novel diffusion model on toric varieties specifically designed for generating conformations of geometrically constrained molecules like protein loops and closed kinematic linkages. The method uses a Jacobian and the R6B6 algorithm to ensure the diffusion process remains on the algebraic variety, avoiding singular states. The model demonstrates good performance on MHC-bound peptide interactions and nanobody loop predictions.

**Strengths:**

The proposed algorithm outperforms the open-source AlphaFold (AF2 and AF3) predictions.

The paper was clearly written, and I deeply appreciate that the authors addressed some limitations in Section E.

**Weaknesses:**

In both experiments in Tab 1 and 2, it seems the experiment has a relatively small scale (<1e3 number of data, <100 test data). And the improvement is relatively small.

**Questions:**

In section E, it seems the algorithm will have trouble when projecting the tangent vector back to the manifold itself. May I ask why you have such difficulty, is it from R6B6 algorithm or from toric variety? I am curious since in Tab. 7 lower right, you use exponential map on torsional space and then there is no such difficulty.

It seems this paper focus on a special type of question: protein generation with a loop. It seems AlphaFold, the main competitor, is a more general model. May I ask if there are other works that specially focuses on this problem (protein with loops)? If nobody did this research before, is it because that the problem is less important than general protein generation problems?

---

> ### Author Response · Authors · 2025-11-24
> **Discussion of weakness and answers to questions**
>
> Discussion of weaknesses: The reviewer is right to observe a limited number of training samples available. In fact, the goal of our work is to address this issue by developing a more data-efficient DL architecture by introducing structural regularization in the form of geometric constraints inherent to loop-like molecular structures. Our improvements, while small, often bring the predictions into X-ray-structure quality regime, which can have dramatic effects for modeling of downstream interactions involving the modeled molecular elements (and, crucially, both MHC-bound peptides and nanobody CDR3 loops are key mediators of such interactions).
>
> Following are the answers to the questions.
>
> Question 1 Answer: Here we are working on algebraic varieties, which are not flat space like torus in which all the movement of the backbone torsions in tangent space can be mapped back to the manifold by exponential map (given that the clashes are ignored). In our context, the proposed noise can result in too large a step which crosses the singular points, so the proposed movement is infeasible and cannot be mapped back, and it is not solvable by R6B6. In [Cao et al. 2023], the authors did extensive testing establishing the efficacy of the R6B6 algorithm which was demonstrated to be more stable and accurate than existing robotic algorithms and have a small closure error 1e-10. If solutions exist, R6B6 can solve the problem and produce all real solutions. Our initial plan was to include more on singularities, but we did not encounter such cases in real examples. We are thinking about designing appropriate singular cases to estimate the performance for such cases in the future. We have added discussions regarding the failure cases and singularity in Appendix E.
>
> Question 2 Answer: We understand and appreciate the reviewer’s concern about this topic. We will answer this question from two aspects. First, loop modeling is a long-time pursuit of computational biologists because of the flexibility in the loop regions and the difficulty in modeling them. A search of the key word ‘protein loop modeling’ in Google Scholar or other search engine, shows works published more than 30 years ago with large numbers of citations. As discussed in [Barozet et al. 2021], loop modeling still poses a challenge to the community and a deep learning method needs to be demonstrated for loop modeling, which calls for the development of methods such as the one presented here. The two examples in the experiments are challenging problems for protein structure prediction and have therapeutic applications as discussed in 'Question 2 Answer' for Reviewer ‘hv3e’ above.
>
> Second, although AF is the state of the art method for structure prediction, the loop regions are often not satisfactory from the predictions of this general model and the diversity of the predicted loops does not match real needs. Our goal is to do local refinement on the loops of the predicted structure and to provide diverse conformations to overcome some existing biases in the AlphaFold model. Given the extraordinary performance of AlphaFold for predicting the protein structures from sequences, any improvement over it would be appreciated.
>
> The main point here is as follows: we start from the prediction of the AlphaFold, and we use the diffusion model to obtain an ensemble of conformations for the loop regions; local refinement and scoring were implemented using AlphaFold, leading to better predictions (in terms of RMSD relative to the experimental structures) and overcoming the bottleneck in AlphaFold method for loops. In AlphaFold, the top-ranking conformations from different random seeds have overall worse performance than the conformations from diffusion.

---

### Official Review · Reviewer_hv3e · 2025-11-01

**Soundness:** 3
**Presentation:** 3
**Contribution:** 3
**Rating:** 6
**Confidence:** 5

**Summary:**

The authors proposed a diffusion model that operates directly on toric varieties to generate protein loop conformations while rigorously enforcing loop closure constraints. Instead of diffusing in Euclidean or unconstrained torsional spaces, the method computes the Jacobian of loop-closure constraints and uses its null space (via SVD) to define valid tangent-space directions for denoising; proposed moves are then projected back to the variety using the R6B6 algebraic loop-closure solver. Architecturally, the score model is SE(3)-equivariant on a heterogeneous graph, and outputs torsional updates expressed in the variety’s tangent basis. Experiments on MHC class I peptides and nanobody CDR3 loops report median RMSD improvements over open-source AlphaFold baselines (≈15–16% for MHC; ≈13% for nanobody CDR3) when selecting top-confidence structures after AF-based refinement. Ablations highlight sensitivity to the maximum noise scale and the number of denoising steps, with ~20 steps and σ_max≈π/10 giving the best trade-off. Limitations include frozen side-chain conformers and occasional step failures near singularities, mitigated by step-size reduction.

**Strengths:**

Casting loop modeling as diffusion on the constraint manifold is a principled advance that bakes physics/kinematics into the generative process, reducing invalid states and sampling waste. Using the Jacobian null space to span feasible motions and R6B6 to project perturbed torsions preserves end constraints exactly, even near stitched-manifold regions with singularities. Consistent RMSD improvements vs. AlphaFold starting structures for two biologically salient tasks (MHC peptides; nanobody CDR3), with clear reporting of medians/means and a simple post-hoc AF scoring pipeline.

**Weaknesses:**

Reliance on AlphaFold pLDDT to pick winners introduces a selection bias and makes it hard to disentangle where the gain truly comes from (diffusion vs. AF rescoring). Freezing side chains risks steric clashes and may undercut accuracy for longer/charged residues; no integrated side-chain generative module is provided. Training/splits emphasize specific loop lengths (e.g., MHC 9–10-mers; CDR3 14–16), and it’s unclear how performance scales to very long loops, multiple loops, or loops with ligand contacts. RMSD is the sole endpoint; missing are physical/energetic plausibility checks (clashes, Ramachandran, rotamer quality) and functional metrics (e.g., MHC binding pose recoveries vs. known anchors).

**Questions:**

1. Beyond RMSD, what biophysical or functional motivations drove the move to toric-variety diffusion (e.g., better capture of anchor residues in MHC, antigen-contact geometries in CDR3)?
2. Why prioritize MHC I and nanobody CDR3 first—were these chosen for availability, difficulty, or expected industrial impact?
3. you choose the six largest components in Δζ to define unknowns—did you compare against conditioning-aware choices (e.g., maximizing the 6×6 submatrix conditioning) to reduce singularity hits?
4. When AF pLDDT is not used for reranking, how do the diffusion ensembles perform under independent quality measures (e.g., MolProbity, clashscore, torsion distributions)?
5. For cases with large improvements, do conformations better reproduce known functional contacts (MHC pockets, CDR3–antigen interfaces), or are gains primarily geometric?

---

> ### Author Response · Authors · 2025-11-24
> **Part I: Discussion of weaknesses and Answers to Question 1-3**
>
> We thank the reviewer for the comments from a biophysical point of view. We provided more analysis as discussed in the answers to the questions below.
> First of all, we would like to briefly discuss the following two points as an answer to weaknesses:  :
> 1. Regarding AF-rescoring, we would like to point out that AF-based scoring is used both in our diffusion protocol and the AF baselines (inherently), with equal number of samples being considered, and thus we assume that the underlying sampling protocol is the key driver of improved results. We have also added an illustration of the difference in sampled conformer distribution (Fig. 7 in Appendix B.5, see our reply to the Question 4 in Part II)
> 2. Regarding side-chain freezing. While the reviewer is correct in that freezing of the side-chains can produce steric clashes, it has been previously shown that given accurate backbone conformation, the side chain torsions can be reconstructed with high accuracy using side-chain-specific modeling tools (see, e.g., https://doi.org/10.1073/pnas.2216438120). With this in mind, we focused on the arguably more difficult problem of handling the protein backbone in the loop region, but retained all atoms of the “frozen” side chains in generated models to make potential side-chain packing easier to implement later.
>
> Following are the answers to the questions.
>
> Question 1 Answer: Our central motivation for developing the approach stems from the poor handling of these inherently important types of molecular structures (see the answer to the next question for more details) by the existing DL-based modeling tools such as AF2 and AF3. We hypothesize that this poor performance of the existing tools is due to high innate conformational diversity of these loop-like structures not being matched by the availability of structural (i.e. X-ray and CryoEM) or evolutionary (multiple sequence alignments) training data. With this assumption, we aim to improve the learning outcomes by reducing the effective dimensionality of the problem by explicitly incorporating the geometric constraints inherent to a closed loop structure into the DL model architecture. Essentially, we are introducing geometric restraints as a form of structural regularization to deal with a data-restricted learning scenario.
>
> Question 2 Answer: We chose these two problems for three main reasons. First of all, these two subproblems of the general protein structure prediction challenge have inherent practical significance: MHC I - peptide complexes provide  a crucial foundation allowing the development of  neoantigen vaccine design, while nanobodies are an important class of therapeutics on their own. More specifically, the loop-like regions we aim to model form critically important interaction interfaces with other proteins: T-Cell Receptors (TCRs) in case of MHC-bound peptides and various target proteins in case of nanobody CDR3 loops. Both systems have a significant amount of research dedicated to them, including the specific structure prediction problems we discuss here. Second, neither of the problems is sufficiently well addressed by the existing structure-modeling tools, including AF2/3, which results in even poorer performance when the downstream interactions involving these loop-like regions need to be modeled. Finally, the third reason is that the likely cause for this reduced performance is the relative paucity of training data representing this highly diverse class of protein structural elements. We hypothesize that in this data-restricted regime, better learning outcomes can be achieved by introducing problem-specific regularization into the model architecture - which we achieve by explicitly incorporating loop-specific geometric constraints.
>
> Question 3 Answer: Combinatorially, there are (n,6) possible selections, checking them one by one is not feasible, so we need a criterion to select the pivots. When mapping the perturbation from tangent space, 6 pivot torsions will change. We want to ‘minimize’ this change. Based on our experiments, the average change is on the scale of 1e-2 for any selection of pivots. When selecting the largest components, the ratio of the change over the perturbation is the smallest, which can minimize the effect of mapping on the direction of the perturbation. If we select the smallest one, the change might completely revert the direction in one dimension of the perturbation, which will distort the proposed noise. So we think the selection of 6 largest components is a robust balance between change and feasibility. We follow the same principle in both training and inference. However, we must  say that exploration of the optimal selection is an open problem and is our future work, especially for the singularity cases.

---

> ### Author Response · Authors · 2025-11-24
> **Part II: Answers to Question 4-5**
>
> Question 4 Answer: We added more discussion after the experiment of CDR3 loops in Section 4.2 regarding diversity. We collected the 100 conformations for one example 8PE1 with large improvement from both AF3 and diffusion model before AF reranking. We next calculated the torsion distribution for the backbone torsions and selected the pair of phi and psi of one residue shown in Fig. 7 in Appendix B.5. As can be seen, the conformations from the diffusion model have better diversity, while the conformations from AF3 have limited diversity. We next examined the MolProbity score and Clashscore using the MolProbity server [Williams et al. 2017]. Because of server limits, we selected 20 structures from AF3 and 20 structures from the diffusion model, and also for the PDB structure. As can be seen in Fig. 8 in Appendix B.5, AF3 can achieve good and similar MolProbity and Clashscore, but does not result in a closer loop region to the PDB structure (the scores of the PDB structure are poor). We have to say that the conformations from the diffusion model do not always score well because we did not take care of the clashes in the current model. We plan to include a function for clashes to guide the movement in the denoising steps in the future to generate more conformations with better quality in terms of Clashscore.
>
> Question 5 Answer: While it is difficult to exactly quantify the functional consequences of model deviations from truth, we can confidently say that for regions involved in intermolecular interactions (such as MHC-bound peptides and CDR3-loops), even small deviations from true structure (such as positioning of side chains)  can have dramatic effects on the ability to model the downstream interaction. As a rule of thumb, structure predictions with <1.0 Angstrom RMSD can be considered to be X-ray-like, while those with <2.0 Angstrom are often considered to be overall successful  [see, e.g. “easy” target definition in https://doi.org/10.7554/eLife.91512]. To better illustrate how RMSD improvements translate to structural differences, we have added illustrations demonstrating examples with large improvement for the MHC and CDR3 dataset, as Fig. 3 and Fig. 4 in Section 4, respectively.

---

### Official Review · Reviewer_Rcnt · 2025-11-01

**Soundness:** 1
**Presentation:** 2
**Contribution:** 2
**Rating:** 2
**Confidence:** 2

**Summary:**

The paper proposes a score‑based diffusion method for loop regions in proteins. The state is parameterized by backbone torsion angles, and loop closure is enforced by solving a 6R inverse‑kinematics (IK) system at every step (R6B6). The forward noise is sampled in the tangent space of the “variety” defined by loop‑closure constraints (obtained as the null space of a 6×n Jacobian), then a “projection” step uses R6B6 to return to the feasible set (Fig. 2; Sec. 3.2–3.3; Alg. 1–2). On MHC‑I peptides (78 cases) and nanobody CDR3 loops (21 cases), the method starts from AlphaFold (AF) structures, generates ensembles for the loop only, refines candidates with AF again, ranks by pLDDT, and reports RMSD improvements vs. AF2/AF3 baselines (Tables 1–2).

**Strengths:**

- **Important problem & plausible inductive bias.** Constraining to torsion angles and enforcing loop closure echoes successful practice in small‑molecule torsional diffusion and docking (Jing et al., 2022; Corso et al., 2023).
- **Operational use of IK (R6B6).** A polynomial IK solver guarantees closed chains when a solution exists (Cao et al., 2023), and the paper documents practical guards (pivot selection; step‑size reduction; ~95% success during inference, Appx. E).
- **Empirical promise.** On two datasets, the best‑of‑ensemble after AF re‑ranking shows median RMSD improvements vs AF2 and AF3 (Tables 1–2).
- **Good connections to classical geometry.** Use of Plücker coordinates / Jacobians aligns with standard robot kinematics practice (e.g., Murray–Li–Sastry).

**Weaknesses:**

1. Misuse/overclaiming of “toric variety.” To call a space toric, one needs a torus action and monomial structure; no such structure is demonstrated. Use “real algebraic subvariety of the $n$‑torus” or similar, unless a toric proof (fan/monomial map) is provided (Cox–Little–Schenck).
2. Tangent space derivation and singularities. Formalize the constraint map $F$ and state assumptions under which IFT yields. Provide a separate treatment near $\mathrm{rank}(P) < 6$. As context, variety‑like conformational spaces can have non‑manifold topology (e.g., cyclo‑octane: union of a sphere and a Klein bottle), underscoring the need to reason about strata and singularities.
3. Projection/retraction properties of R6B6 step. If R6B6 is used as a “projection,” give (local) existence/uniqueness, continuity, and ideally differentiability results, or at least an error bound showing it behaves as a retraction for small steps (see Absil–Mahony–Sepulchre, Ch. 4). Today’s text states “closest solution” but no metric on periodic angles or tie‑breaking provides uniqueness.
4. Score/forward mismatch. Either (a) redefine the forward kernel as the pushforward through the R6B6 map and train the correct score, or (b) justify a small‑step approximation providing provable bounds that the Gaussian‑tangent score suffices. Without this, the reverse dynamics are not those of the stated forward process (Song et al., 2021).
5. Sampling rule lacks derivation. Provide a derivation of Alg. 2’s $\Delta \tau$ update from a particular SDE/ODE (VE or VP) and show training–sampling compatibility (cf. Song et al., 2021; EDM design rules).
6. Baselines not protocol‑matched. Please include AF2/AF3 ensemble baselines with equivalent number of candidates, refinement, and ranking (AF can stochastically sample diverse states; Del Álamo et al., 2022). Also report statistical tests / CIs for Tables 1–2, and ablations (no R6B6, no AF re‑refinement, different pivot criteria).

**Questions:**

1. Toric or not? Can you prove the loop‑closure set is a toric variety (torus action, monomial parametrization/fan), or will you revise the terminology? (Sec. 3.1).
2. Tangent space rigor. Please define $F$ and prove $DF =P$, then state IFT conditions under which $T_x mathcal{V} = \mathrm{ker}P(x)$ holds; clarify the argument under Eq. (1)–(2) (p. 5).
3. “Projection” properties. For the R6B6 step, can you show local single‑valuedness, continuity, and/or a first‑order accuracy bound making it a (local) retraction? If not, please avoid “projection” terminology or define the metric.
4. Forward/score consistency. How do you reconcile training on the Gaussian tangent score with sampling that uses the R6B6‑transformed perturbation? Any theorem or bound (small‑step limit) ensuring equivalence? (Alg. 1–2).
5. Sampling update derivation. From which SDE/ODE is $g(t)$ derived (VE or VP), and how do you ensure consistency with the training loss? Please provide the derivation in the appendix.
6. Protocol‑matched baselines. Please include AF ensemble sampling (e.g., altered MSA depth, dropout) with the same number of candidates and AF re‑refinement + ranking, and report paired tests/intervals.

---

> ### Author Response · Authors · 2025-11-24
> **Part I: Answers to Question 1-3**
>
> We appreciate the review comments from an algebraic geometry perspective. These questions let us rethink the missing points in our work and led to substantial improvements. The following are answers to the questions.
>
> Question 1 Answer: Thank you for pointing this out. Indeed we used the word loosely, motivated from the open chain that lives on the n-Torus. In general, a protein loop is composed of a sequence of amino acids. In the loop, the backbone torsion angles (phi, psi) range from –pi to pi and are inherently circular, and live on the n-torus if we neglect steric clashes. However, these torsions have to move in a concerted way to maintain the loop closure conditions at two ends of the loop. Based on the closure conditions, we can formulate polynomial equations, and the zero set of these polynomial equations forms an algebraic variety. So we mislabeled the algebraic variety with torus parameters as toric variety. In any case, our work does not depend on the toric property. We have corrected our term to ‘real algebraic subvariety of n-torus’ and modified the terminology accordingly in the Section 3. We also changed the title of the paper from “Toric” to “Algebraic”. Hopefully this addresses the reviewer’s concerns. All the modified words in the revised manuscript are in red color.
>
> Question 2 Answer: We provided a reference to the IFT in the form we use it in the book [Sommese & Wampler, 2005]. Typical singularities one encounters are “elbows” where one of the n-6 driving torsions exceeds the range for which a solution may exist, typically at such values two distinct poses merge and disappear or situations where different solution branches cross, as in cyclooctane. At such points the Jacobian involving the given set of pivot axes vanishes. The statement invoking the IFT simply refers to the fact that the system of polynomial equations is differentiable with respect to all the torsion angle half-tangents and the IFT guarantees the existence of a nearby solution provided the Jacobian is of full rank. In our case, the coefficients of the polynomial system we solve are locally differentiable functions of the driving torsions. This statement requires some work to fully establish but essentially it says that the length of a chain of rotors and the orientation of its end frame is a differentiable function of the chain torsions. Although we solve a standard system of polynomials to close the loop, we follow the common procedure of computing the Jacobian directly from the Plucker coordinates of the pivot axes of the reconstructed chain.  The Jacobian is defined based on the current point on the variety and we choose 6 pivots and then the determinant of the corresponding 6x6 submatrix can be calculated to decide independence. In our case, determining the precise image of the map requires the identification of the branch of the variety that we construct as the inverse image of zero to reconstruct the neighborhood of the given loop rather than that of a different realization. For that a proximity test is essential. We modified the sentences under equation 1-2, stating that.
>
> Question 3 Answer: We agree that ‘projection’ might not be an appropriate term here. To be consistent with the exponential map, we changed from ‘projection’ to ‘map’. We clarified the definition of ‘closest solution’ and added the metric to determine that in the revised manuscript: the closest solution $\zeta_i^{p}$ to the original 6 torsion values $\zeta_i^{o}$ is selected to have the smallest metric $\sum_{i=1}^6 |\zeta_i^{p} - \zeta_i^{o}|$, where $|\cdot|=2\pi - |\cdot|$, if $|\cdot|>\pi$. In the current work (also in [Jing et al., 2022; Corso et al., 2023]), we are exploring the variety without searching for optimal directions. Employing R6B6 to map from the tangent space to the variety we typically incur 6 pivot torsion differences in the order of 1e-2 on average, while all other torsions are kept fixed.

---

> > ### Author Response · Authors · 2025-11-24
> > **Part II: Answers to Question 4-5**
> >
> > Question 4 Answer: Introducing noise in the tangent space is natural and it avoids creating test structures with abrupt breaks that can make the learning process meaningless. We carefully follow the consistency in the forward and reverse processes to make the learning meaningful for our work. In the training, the Gaussian tangent noise has n-6 orthogonal dimensions. Combining this noise with n-6 orthonormal tangent vectors provides the perturbation to the torsions in the loop. R6B6 was used to close the loop (mathematically speaking, it maps the tangential step back to the variety), from which the neural network can learn from a realistic structure. In the sampling phase, the network first predicts n torsions in the torsional layer and this n-dimensional vector is orthogonally projected to the tangent space in the magnitude layer by multiplying with the n-6 orthonormal basis vectors of the tangent space (n-6 dimensions), which produces the Gaussian tangent score (n-6 dimensions) at this step. Multiplying the Gaussian score with basis vectors, we obtain the perturbation to the loop in the tangent space and then use R6B6 to map the movement in tangent space to the variety which keeps the loop closed. We did not directly modify the Gaussian tangent score using R6B6. When encountering failure steps, the step size will be reduced by 10%. In this sense, the score was reduced by 10% but not by R6B6. Moreover, based on our testing, the map introduced by R6B6 is not providing much change (on average at the scale 1e-2) to the torsions. We have to point out that only 6 torsions (Pivot torsions) were changed at the scale 1e-2 and all the remaining n-6 (Driver) torsions were set at the exact same values from the multiplication of the Gaussian score from the neural network and the basis vectors.
> >
> > Question 5 Answer: Without interrupting the section numbering in the previous version of the manuscript, we added the derivation in Appendix F. The model is derived using VE SDE. Regarding consistency, the trained score model provides the guidance (learned by the score function) in the reverse diffusion process. The movement at each denoising step is determined by the score estimate, which moves the current point towards the desired data distribution. The neural network was trained with a loss function to accurately model the gradients of the true data distribution. The reverse process, which uses the output from the neural network to guide the steps, is consistent with the training objective. Usually the smaller training loss corresponds to a more accurate score function, which will lead to better sampling. However, overfitting of the trained model is always a problem we need to safeguard against. In the experiments, we first trained the model with 100, 200, 300, 400, 500 epochs (the training loss is decreasing as the number of epochs increases) and then generated ensembles using the trained models. Although all these 5 models produced ensembles with similar variation, the generated conformations from the model trained with 500 epochs did not have the best performance as compared to the model with 300 epochs. Since we are learning from the ground-truth structures from PDB, the closer results to the structures in PDB indicate a better model even though the training loss was not the smallest. Consequently, we used the model with 300 epochs to ensure consistency.

---

> ### Author Response · Authors · 2025-11-24
> **Part III: Answer to Question 6**
>
> Question 6 Answer: When comparing our protocol to AF2 and AF3 baselines (Table 1 and Table 2), we used the same number of conformations for the diffusion-based protocol and the baseline: 20 from AF2 against 20 from diffusion model for MHC I, 100 from AF3 against 100 from diffusion model in CDR3 loops. The statistical tests for all cases are included as histograms in the Appendix B.3 in the revised manuscript. For AF2, we used the ColabFold implementation of AlphaFold2 with ColabDB/MMseqs Multiple sequence alignments, and ran each of the 5 standard parameter sets with either 4 (MHC I) or 20 (CDR3 loops) random seeds, for a total of 20 or 100 samples, respectively. In a similar manner, we ran the standard implementation of AF3 using standard MSAs to generate a total of 20 (MHC I) or 100 (CDR3 Loops) samples using different random seeds. Dropout was not used. The relevant information was  added as “Appendix H: Alphafold2/3 baselines”. Regarding refinement and ranking, we assume that comparing AF-refined diffusion samples to AF-derived samples without refinement is fair as the latter implicitly incorporate AF refinement a scoring in a certain sense.
>
> Regarding ablations: For the model without R6B6, the generated loop will be meaningless because the loop won’t be closed and the generated samples will have broken bonds. With respect to non-refined diffusion samples, while we haven’t been able to fully address the reviewer’s concerns, we have conducted a limited study of conformational diversity of non-refined samples (see Fig 7 and Fig 8 in Appendix B.5). We thank the reviewer for the ideas and will try to do the suggested ablations as a part of the future work.

---

### Meta-Review · Area_Chair_Y3fV · 2026-01-04

**Summary:**

This paper proposes a diffusion model that operates directly on toric varieties to generate protein loop conformations while rigorously enforcing loop closure constraints. The reviews are quite diverse. Positive reviewers acknowledge the novelty and contribution, while negative reviewers pointed concerns in the experiments. I agree with Reviewer mnqi that "If they prefer to keep this as an empirical paper, they must compensate with additional experiments demonstrating robustness.", which seems not easy to address in revisions.

**Reviewer Concerns:**

Most concerns are well addressed by the rebuttal.

**Reviewer Scores:**

I appreciate the authors' efforts to thoroughly address the rebuttal concerns, and I think some reviewers would increase their scores. Nevertheless, the expected scores still do not meet the ICLR acceptance threshold.

---

### Decision · Program_Chairs · 2026-01-26

Reject